# Episodic evolution of coadapted sets of amino acid sites in mitochondrial proteins

**Alexey D. Neverov** [1] *, **Anfisa V. Popova** [1], **Gennady G. Fedonin** [1,2,3], **Evgeny A. Cheremukhin** [4], **Galya V. Klink** [2], **Georgii A. Bazykin** [2,5]

1 Department of Molecular Diagnostics, Central Research Institute for Epidemiology, Moscow, Russia, 2 Institute for Information Transmission Problems (Kharkevich Institute), Russian Academy of Sciences, Moscow, Russia, 3 Moscow Institute of Physics and Technology, Dolgoprudny, Moscow region, Russia, 4 Department of Chemistry, M. V. Lomonosov Moscow State University, Moscow, Russia, 5 Skolkovo Institute of Science and Technology, Skolkovo, Russia

* neva_2000@mail.ru

## Abstract

The rate of evolution differs between protein sites and changes with time. However, the link between these two phenomena remains poorly understood. Here, we design a phylogenetic approach for distinguishing pairs of amino acid sites that evolve concordantly, i.e., such that substitutions at one site trigger subsequent substitutions at the other; and also pairs of sites that evolve discordantly, so that substitutions at one site impede subsequent substitutions at the other. We distinguish groups of amino acid sites that undergo coordinated evolution and evolve discordantly from other such groups. In mitochondrion-encoded proteins of metazoans and fungi, we show that concordantly evolving sites are clustered in protein structures. By analysing the phylogenetic patterns of substitutions at concordantly and discordantly evolving site pairs, we find that concordant evolution has two distinct causes: epistatic interactions between amino acid substitutions and episodes of selection independently affecting substitutions at different sites. The rate of substitutions at concordantly evolving groups of protein sites changes in the course of evolution, indicating episodes of selection limited to some of the lineages. The phylogenetic positions of these changes are consistent between proteins, suggesting common selective forces underlying them.

## Author summary

The mode and rate of evolution of a protein site depends on the effect of its mutations on protein fitness. The fitness effect of a mutation itself can change in the course of evolution for at least two reasons. First, it can be modulated by substitutions occurring at other sites, a phenomenon called epistasis. Second, changes in selection can be non-epistatic, affecting sites independently of one another. Here, we analyse substitutions accumulated by the evolving lineages of the five proteins encoded by the mitochondrial genomes of thousands of species of metazoans and fungi. We show that substitutions at different amino acid sites occur in a coordinated fashion, and this coordination is caused both by epistasis and by episodes of selection affecting groups of sites. We partition each protein into several

and all data and codes are available under GPL 3.0 license: https://github.com/gFedonin/EpiStat

**Funding:** GVK and GAB were partially supported by the Molecular and Cellular Biology Program of the Russian Academy of Sciences. The funders had no role in study design, data collection and analysis, decision to publish, or preparation of the manuscript.

**Competing interests:** The authors have declared that no competing interests exist.

groups of concordantly evolving sites such that evolution of sites from different groups is discordant, and show that the proteins encoded by the mitochondrial genome consist of coevolving structural blocks. Some of these blocks have a clear functional specialization, e.g. are associated with interfaces between proteins composing respiratory complexes. Together, our results reveal a previously unrecognized complexity in the causes of variation in evolutionary rates between protein sites.

## Introduction

### Correlated occurrence of amino acids at different sites

The rate at which individual protein sites accumulate substitutions changes in the course of evolution, which violates the assumptions of evolutionary models and may cause problems for phylogenetic reconstruction. This variability can uniformly affect all substitution types ("heterotachy" [1,2]) or differentiate between them ("heteropecilly" [2]). The substitution rate is the product of the rate at which mutations arise and the rate at which they are fixed [3], and can be affected by changes in either of these processes. The fixation probability of a mutation is affected by several factors, namely, changes in mutation rate, in effective population size due to demographic processes, or in selection favoring some variants over others. The direction or magnitude of selection at a site itself can change due to multiple forces including changes in environmentally induced constraints or substitutions at other epistatically interacting genomic sites.

Changes in selection appear to play a major role [4,5]. One type of evidence for this is the correlations between the occurrence of different amino acids at pairs of sites in multiple alignments (MSAs) of homologous sequences. Such correlations, inferred using direct coupling analysis (DCA) or related methods, are associated with physical proximity, and are sufficiently strong that they can be used to infer protein structures and interprotein contacts [6–12] and to predict fitness effects of substitutions [10,13,14].

Given the success of these approaches, it is tempting to aggregate cooccurrence data across many sites to get a bird's eye view of the constraints on the evolution of the entire protein. Several computational approaches to this have been proposed, revealing the partitioning of protein sites into dense coevolving domains spatially separated on the structure and associated with biological functions [15–18]. Modelling indicates that such coevolving domains may arise naturally, e.g. in the elastic network models of allosteric proteins [19–21].

### Complexity of mitochondrial evolution

Mitochondrial-encoded proteins are a good model system for coevolution between sites. On the one hand, sequencing data is abundant across all eukaryotes, and structures and functions of proteins and protein complexes are well understood. In particular, COX1 is a universal genetic barcode and serves as a key genetic marker for unraveling taxonomic relationships in animals [22,23].

On the other hand, mitochondrial evolution is a complex process. This complexity has been mainly studied from the viewpoint of phylogenetic reconstruction. Mitochondrial proteins violate the basic assumptions of phylogenetic methods, namely, homogeneity of the processes of amino acid substitutions along lineages, between sites in alignment and between character states within sites [24]. This variation can arise due to differences in mutation [25,26] and/or selection [27,28]. Together with the applied importance of mitochondrial markers, this has

motivated development of approaches that relax assumptions of simpler evolutionary models, allowing for heterogeneity between sites [29–31] or between lineages [1,2,25] and thus improving accuracy of phylogenetic reconstructions.

A remarkable example of heterogeneity of substitution rates due to variations of selective constraints is provided by mitochondrial proteins COX1, COX2 and COX3 of the cytochrome oxidase complex (COX). Sites at the surfaces of these proteins that are involved in contact interfaces with other proteins encoded by mitochondrial or nuclear genomes evolve at systematically different rates, compared to sites not involved in such interfaces [32]. The direction of this difference varies among proteins. Evolution is decelerated, indicating stronger purifying selection, at contact interface sites of COX2 and COX3, compared to exposed noncontact non-interface sites. By contrast, in COX1, selective constraint is stronger at non-interface sites, possibly due to their involvement in formation of heme environment. In all three proteins, sites in contact with other mitochondrial-encoded proteins evolve more slowly than those in contact with nuclear-encoded proteins [32].

## Epistasis in mitochondrial proteins

Several lines of evidence suggest that fitness conferred by different amino acids at an amino acid site of a mitochondrial protein depends on the amino acids found at other sites of the same mitochondrially encoded protein, or of other proteins encoded by mitochondrial or nuclear genomes—i.e., that these sites are involved in epistatic interactions [33]. First, epistasis has been inferred from compensated pathogenic deviations [34], i.e., cases when a human pathogenic variant has been observed in wildtype in some species. Several such cases have been described for the mitochondrion-encoded proteins of oxidative phosphorylation (OXPHOS). Detailed studies have shown that such variants can be neutralized by substitutions at other sites proximal in the 3D structure or in the same interaction interface [35]. Substitutions in OXPHOS proteins with predicted strong deleterious effects in humans arose repeatedly on the phylogeny of mammals, birds and reptiles. Many of these substitutions (37%) were rapidly compensated on the same tree branches by substitutions at other sites in contact with them on the protein structures [36] the rest had no clear compensating counterpart.

Second, high prevalence of epistasis has been shown in experiments. Replacements of mitochondria in genetically divergent strains of *Saccharomyces cerevisiae* yeast revealed a strong dependence of growth rates on the mitonuclear interactions [37]. In mice, epistasis with a polymorphism in mitochondrially encoded CYTB aggravates the effect of a pathogenic mutation of nuclearly encoded Bcs1l [38].

Finally, sites in close proximity in protein structures tend to coevolve, and this is most plausibly explained by epistatic interactions among contacting sites. This has been observed in COX1 [39] and COX2 [40] proteins as well as in mitochondria-nuclear interfaces [41–43]. Still, our understanding of interactions between sites and the role of such interactions in the evolution of OXPHOS proteins remains limited.

Here, extending our previous work [44,45], we develop a phylogenetic method for inference of protein sites involved in concordant or discordant evolution. Roughly, for each pair of sites, we count the number of cases when a substitution at one of these sites rapidly follows a substitution in the other within the same evolutionary lineage. An excess of such cases is suggestive of positive epistasis, whereby the first substitution increases the fitness gain associated with the second substitution; while their deficit is suggestive of negative epistasis, whereby the first substitution decreases the selective advantage, or increases the deleterious effect, of the second one. To address this formally, we compare this number to that expected if substitutions at each site proceed independently of each other. This is done by calculating the association statistic,

which is positive if the number of pairs of consecutive substitutions at these sites is unexpectedly high, and negative, if it is unexpectedly low. We apply this method to the evolution of five proteins COX1, COX2, COX3, CYTB and ATP6 encoded in mitochondrial genomes of metazoans and fungi.

In all proteins, we observe many site pairs with excess or deficit of such rapid pairs of consecutive substitutions, which we further refer to as concordantly and discordantly evolving site pairs. By analyzing substitutions at different lineages at these site pairs, we then show that these biases are partially caused by epistatic interactions as per our hypothesis, while partially they are most consistent with additive episodic selection. Using modularity, a community detection method in networks with positive and negative links, we partition sites in each protein into coevolving groups with a high concordance of evolution within and discordance between groups. Sites within a group tend to be located densely on the protein structure. We show that groups distinguished on the basis of concordance of substitutions at individual site pairs also demonstrate coincident changes in substitution rates: these changes occur in concert between all five mitochondrially encoded OXPHOS proteins, suggesting that they have a common cause.

## Results

### Concordantly and discordantly evolving pairs of sites

First, we modified the previously developed phylogenetic approach [44] to detect pairs of concordantly evolving amino acid sites in mitochondrion-encoded proteins (Fig 1). In brief, we reconstructed the phylogenetic positions of all substitutions, and identified pairs of sites such that a substitution at one of them was frequently rapidly followed by a subsequent substitution in the other, as evidenced by higher than expected values of the epistatic statistic [44]. In each of the five studied proteins, we observed strong positive associations of substitutions for some of the site pairs that was significantly above that expected randomly (Table 1 and S1 Fig and S1 Data). Surprisingly, at a number of other site pairs, the observed epistatic statistics were significantly lower than expected. This indicates that a substitution at one of these sites was followed by a substitution at the other site more rarely than expected randomly, suggesting discordant evolution of these sites (Fig 1). Statistically, the signal of discordant evolution was much stronger than that of concordant evolution, in terms of larger number of significant site pairs for FDR<0.3 (Table 1 and S1 and S2 Figs and S1 and S2 Spreadsheets), although a direct

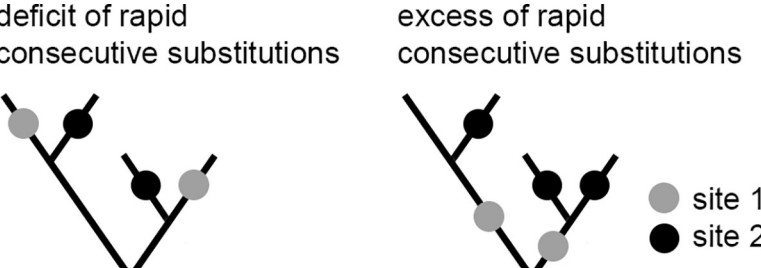

**Fig 1. Phylogenetic approach for detection of concordantly and discordantly evolving site pairs.** The approach is based on counting the rapid consecutive substitutions at site pairs. Two substitutions at sites 1 (gray dots) and 2 (black dots) are consecutive if they occur in the same line of descent (i.e., one is ancestral to the other) and no other substitutions at these sites occurred over the time period between them. A deficit of rapid consecutive substitutions implies discordant evolution of these two sites (here, 0 pairs of consecutive substitutions; left), while an excess of such substitutions implies their concordant evolution (here, 3 pairs of consecutive substitutions; right).

**Table 1. Numbers of concordantly (+) and discordantly (-) evolving site pairs in mitochondrial-encoded proteins.**

| gene | sign of association statistics | #pairs | nominal p-value threshold for FDR<30% |
|---|---|---|---|
| ATP6 | | | |
| | + | 2106 | 0.0262 |
| | - | 6440 | 0.05 |
| CYTB | | | |
| | + | 2452 | 0.0099 |
| | - | 14354 | 0.05 |
| COX1 | | | |
| | + | 12309 | 0.0316 |
| | - | 13663 | 0.05 |
| COX2 | | | |
| | + | 1086 | 0.0135 |
| | - | 4636 | 0.05 |
| COX3 | | | |
| | + | 3809 | 0.0368 |
| | - | 4882 | 0.05 |

For each gene, the number of predicted site pairs (#pairs) and the nominal p-values corresponding to the false discovery rate (FDR) <0.3 (see Methods) are shown.

comparison is impossible because the test has different power to detect positive and negative associations.

Conceivably, biases in phylogenetic distribution of substitutions, and specifically hemiplasies (spurious convergent or parallel events), could arise from errors in phylogenetic reconstruction. To control for this, we devised a procedure for accounting for uncertainty of phylogenetic reconstruction. Under this procedure, we weight all potentially hemiplasic substitutions corresponding to a single actual substitution by the reciprocal of the number of such hemiplasic substitutions, so that their contribution to the statistic is not inflated (see Methods). Applying this procedure slightly decreased the power of the test, reducing the number of inferred site pairs with FDR<0.3 for each gene (S1 Table). However, this correction changed the list of concordantly and discordantly evolving site pairs only slightly; e.g., for COX2, 84 and 77 of the top 100 concordantly and discordantly evolved site pairs coincided between the two lists.

## Concordant or discordant evolution do not result from site-specific uncorrelated episodes of positive selection

Conceivably, the observed concordance and discordance of evolution between site pairs could originate from non-uniform substitution rate within individual sites. Such non-uniformity could result, for example, from episodes of positive selection, clonal interference between non-recombining mitochondrial genomes, and/or genetic hitchhiking with advantageous mutations.

We tested our method to avoid false positive findings in simulated evolution of independent sites in presence of positive selection, using two different approaches.

Firstly, we simulated independent episodic positive selection at individual sites. For this, we used the SELVA phylogenetic simulator [46]. It allows to model amino acid evolution of a site in a set of evolving lineages along a predefined phylogeny while varying the relative fitness corresponding to different amino acids at this site. We thus varied these amino acid preferences at each site, which triggered recurrent episodes of positive selection causing adaptive

substitutions. Nevertheless, in the resulting dataset of concatenated SELVA-simulated sites, no signal of concordant or discordant evolution was observed: no site pairs corresponding to FDR<0.3 were predicted (S3A and S3C Fig and S3 Spreadsheet).

Secondly, we modeled linked evolution of multiple sites under extensive clonal interference. For this, we used SantaSim, a forward-time simulator of molecular evolution in a population [47], modelling a gene consisting of multiple amino acid sites in each of which one of the amino acids was favored. The starting genotype was picked at random, allowing multiple beneficial mutations and therefore multiple adaptive pathways causing extensive clonal interference and hitchhiking. Indeed, starting from a relatively unfit genotype, the population segregated into multiple competing clones represented on the phylogenetic tree by multiple long-living clades (S4 Fig). The resulting sequences were used for phylogenetic reconstruction, and the signal of concordant or discordant evolution was measured; again, no signal of concordant evolution was observed, and only a single discordantly evolved site pair was detected (S3B and S3D Fig and S3 Spreadsheet).

## Disentangling direct and indirect associations between sites

In order to be able to compare the strength of concordance or discordance between different site pairs, we converted the epistatic statistic into a normalized form. For this, the statistic for each site pair was z-score transformed, and the resulting z-scores across all site pairs were divided by their maximum value. The resulting values are referred to below as pseudo-correlations; they fall into the range between -1 and 1, with positive values corresponding to concordant evolution, negative values, to discordant evolution, and 0, to independent evolution of sites.

To single out the direct causative correlations against the background of indirect ones, following previous studies [7], we then defined the association statistics as a causative measure of concordant or discordant evolution. For site pairs with positive pseudo-correlations, the association statistic was set to equal the values of the corresponding partial correlations if these correlations were positive, or to zero otherwise. For site pairs with negative pseudo-correlations, we set association statistics to equal to pseudo-correlations themselves. We ranked site pairs according to the values of the association statistics.

## Our approach is able to detect truly positively and negatively epistatically evolved site pairs in idealized conditions of in-silico evolutionary experiments

Positive epistatic interactions between substitutions in a site pair are expected to lead to a signal of concordant evolution for this site pair, and negative epistatic interactions, to a signal of discordant evolution. To illustrate this, we simulated in-silico evolution of a genome that contained some epistatically interacting site pairs using the MimicrEE2 forward evolutionary simulator [48]. Each site of each genotype contained one of the two possible alleles, "a" or "A". We studied the evolution of the genome of the 100 site length which contained ten pairs of sites in positive epistasis and ten pairs of sites in negative epistasis; the remaining sites evolved neutrally. The two-positional fitness landscapes in all positive epistatic site pairs were identical, and the two-positional fitness landscapes in all negative epistatic site pairs were also identical. All epistasis was pairwise, i.e., the fitness values of alleles in a pair of epistatically interacting sites were independent of alleles in other sites. As a negative control, we modelled evolution of a genome containing only neutrally evolved sites.

This simulation is simplistic, and unrealistic in several respects. In particular, we assume that all selection is epistatic, and non-epistatically interacting sites are neutral; that simulations

start from the most fit genotype; and that selection acting at the considered sites shapes the phylogeny. Still, it illustrates that the signal of epistasis is picked up by our approach. As expected, positive epistatic interactions between evolving sites elevated the values of the association statistics for these sites, and negative interactions, decreased these values. The top eight site pairs with the highest positive values of association statistics were true positively epistatically interacting site pairs (S5 and S6 Figs and S4 Spreadsheet). Of note, just one epistatically interacting site pair was among the top ten predicted pairs in the same list ordered according to pseudo-correlations, emphasizing the need for disentangling direct interactions from indirect ones. Five site pairs among the ten with the highest absolute values of negative association statistics were true negative epistatically interacting site pairs.

In the negative control where all sites in the genome were neutrally evolving, we did not observe any significantly concordantly or discordantly evolved site pairs for FDR<0.3.

## Concordant and discordant evolution can be caused by episodes of epistatic or additive selection

The observed concordance and discordance of evolution may result from epistatic interactions between sites within a pair. In addition, it could result from non-epistatic episodic selection affecting both these sites, such that these sites are selected only part of the time but substitutions in them contribute to fitness additively. To discriminate between these scenarios, we note that epistatic and additive episodic evolution can be distinguished by the phylogenetic patterns they cause. With regard to concordant evolution, episodic selection, e.g. acting within a certain clade, is expected to bias the substitution patterns at all sites affected by this selection within this entire clade, independent of whether these substitutions occur in the same or in different lineages (Fig 2, I). By contrast, positive epistasis is only expected to lead to an excess of rapid consecutive substitutions, and is not expected to bias the phylogenetic distribution of substitutions that do not fall into the same lineage (Fig 2, IV). With regard to discordant evolution, if it is caused by distinct episodes of non-epistatic selection between sites of a pair, we expect to observe "repulsion" of substitutions not only within lineages but also between them (Fig 2, III). Conversely, if discordance is caused by negative epistasis, we expect a deficit of rapid consecutive substitutions, while no bias is expected for substitutions in different lineages (such substitutions may even be phylogenetically clustered if negative epistasis is accompanied by concurrent episodic selection) (Fig 2, II). In general, additive episodic selection is expected to affect pairs of substitutions equally independent of whether they are consecutive or not; while epistatic selection is expected to only affect consecutive pairs.

To measure whether substitutions in different lineages tend to cluster on the tree, we calculated the average distances between all pairs of nonconsecutive substitutions for each pair of sites, and defined the clustering z-score as the normalized difference between the expected and observed distances.

We found that pairs of significantly concordantly and discordantly evolved protein sites differed both in whether there was an excess or deficit of rapid consecutive substitutions, and whether there was a clustering or repulsion of non-consecutive ones (Fig 3). Substitution patterns at some site pairs were indicative of epistatic interactions between them (Figs 4A and 5A), while at others, they suggested non-epistatic episodic selection (Figs 4B and 5B). We examined the correlations across site pairs between the association statistics measuring the excess (deficit) of rapid consecutive substitutions and the clustering z-score statistics measuring phylogenetic clustering (repulsion) of nonconsecutive ones, and found that the prevailing direction of association between these measures differed between proteins. The observed patterns allowed us to classify the proteins by the inferred cause of the observed concordance or

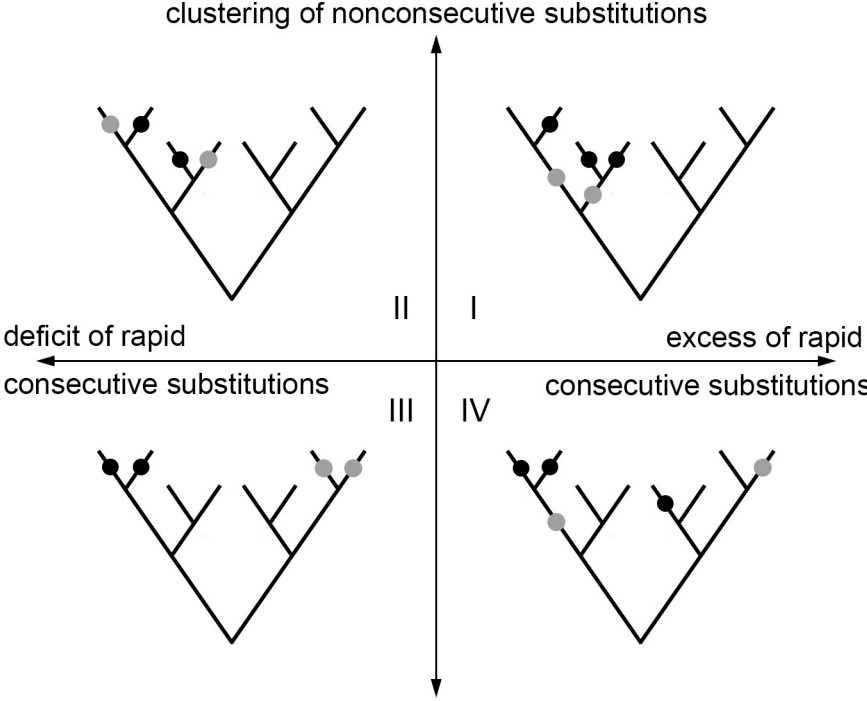

**Fig 2. Distinguishing between additive and epistatic episodic selection.** Each pair of sites can be characterized by excess or deficit of rapid substitutions in the same lineage (horizontal axis), vs. clustering or repulsion of substitutions in different lineages (vertical axis). On the scheme, substitutions at one of the two considered sites are shown in black, and substitutions at the other are shown in gray. An excess of rapid consecutive substitutions accompanied by clustering of nonconsecutive substitutions (quadrant I) implies concurrent episodic selection and can be observed without any epistatic interactions between sites. Analogously, a deficit of rapid consecutive substitutions accompanied by repulsion of nonconsecutive substitutions (quadrant III) implies episodes of selection that are distinct between the two sites and does not necessarily require epistasis. By contrast, an excess of rapid consecutive substitutions together with repulsion of nonconsecutive substitutions (quadrant IV) cannot be explained by episodic selection alone and is indicative of positive epistasis. Similarly, a deficit of rapid consecutive substitutions along with clustering of nonconsecutive ones indicates negative epistasis.

discordance (Fig 3 and S2 Table). In ATP6, CYTB, COX1 and COX3, the two statistics are positively correlated, so that concordantly evolving site pairs tend to display "clustering" of nonconsecutive substitutions, implying that additive episodic selection contributes to the signal of concordance. However, positive epistasis is also present: in all proteins, many site pairs with high association statistics (>0.03) display nonpositive clustering z-scores for nonconsecutive substitutions, which means that the observed concordance does not result from episodic selection (Fig 3A). Discordantly evolving pairs of ATP6 tend to also display repulsion of nonconsecutive substitutions, which again implies additive episodic selection that alternates between sites (Fig 3B).

However, epistasis needs to be invoked to explain the patterns observed in discordantly evolving pairs of COX1, COX2 and COX3. In these genes, the strength of clustering of nonconsecutive substitutions and the deficit of rapid consecutive ones are negatively correlated. In other words, the site pairs at which non-consecutive substitutions are located at nearby phylogenetic branches also tend to be those where consecutive substitutions are most underrepresented (Fig 3B). This observation cannot be explained by additive episodic selection alone, since clustering of substitutions due to an episode of selection within a clade can be expected to lead to concordant, rather than discordant, evolution within that clade. To explain it, one

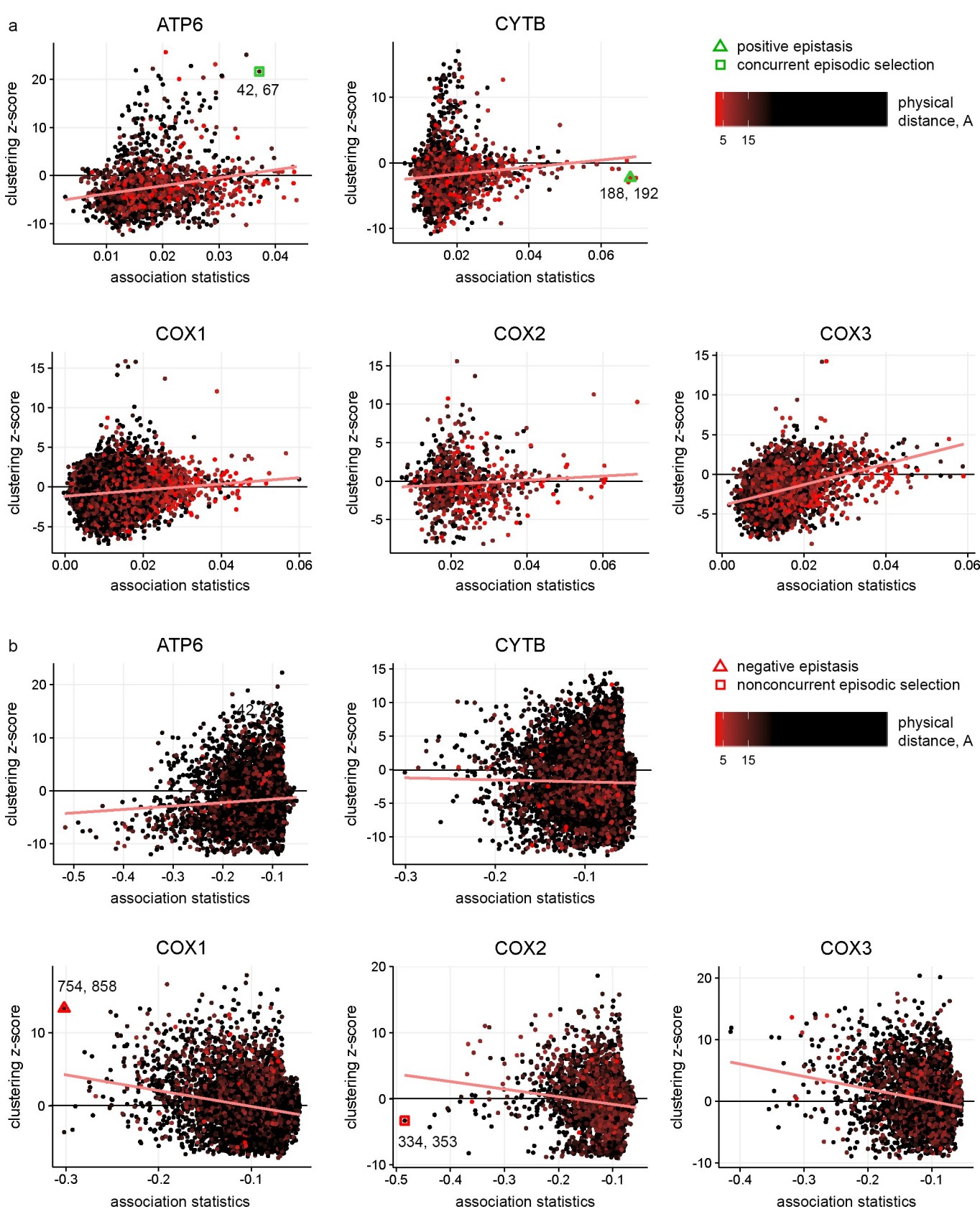

**Fig 3. Relationships between the excess (deficit) of rapid consecutive substitutions and clustering (repulsion) of non-consecutive substitutions distinguish between epistasis and additive episodic selection.** Each dot represents a site pair with a significant excess (strong positive pseudo-correlation) or deficit (strong negative pseudo-correlation) of rapid consecutive substitutions. The vertical axis indicates the excess of clustering (high positive values of clustering z-score) or repulsion (low negative values of clustering z-score) of non-consecutive substitutions for this site pair. Dot color corresponds to the distance between the sites in 3D protein structures: red for contacting sites and black for distant sites. Pairs of colocalized sites in protein structures are more common among concordantly evolved site pairs with strong positive pseudo-correlations than among those with weak or negative pseudo-correlations. The trends for site pairs with positive and negative pseudo-correlations are indicated with red lines. Marked are site pairs representative of positive (CYTB) and negative (COX1) epistasis with no signs of episodic selection, or of concordant (ATP6) or discordant (COX2) evolution likely resulting from additive episodic selection; these sites are also presented in Figs 4 and 5; the numbers indicate the amino acid sites for the corresponding pairs.

needs to invoke an episode of selection favoring substitutions at both sites individually, but disfavouring their combination–i.e., an episode of negatively epistatic selection. For CYTB, there is no significant correlation between association statistics and clustering z-scores; however, site pairs with high absolute values of association statistics ($>0.2$) mostly have positive values of clustering z-scores, implying that the deficit of rapid consecutive substitutions at these site pairs is not due to repulsion of all substitutions, and results from negative epistasis (Fig 3B).

## Concordant evolution is associated with proximity in protein structures

We asked how concordantly and discordantly evolving site pairs are positioned relative to each other in 3D protein structures. For significantly concordantly evolving site pairs (those with positive pseudo-correlations and significant epistatic statistics, see Methods), we found that higher values of the association statistic and higher pseudo-correlations are associated with smaller distances between sites (Tables 2 and S3). By contrast, for two of the proteins, COX2 and COX3, stronger significant negative association statistics are characteristic of sites remote in the 3D structure. For discordantly evolved site pairs, for three other proteins, ATP6, COX1 and CYTB, no significant associations with distances on structures were found (Table 2). To better understand the association between concordance and spatial proximity, we directly compared the 3D distances between sites in concordantly and discordantly evolving site pairs. For all five proteins, the concordant sites were closer to each other than the discordant ones (Mann Whitney test, $P<10^{-16}$, S4 Table and Fig 3 and S5 Spreadsheet). For each gene, the fraction of contacts among concordantly evolving pairs of sites was higher than expected (Binomial test $P<2.2E-16$), and among discordantly evolving ones, lower than expected (Binomial test $P = 1.614e-12$ for ATP6, $< 2.2e-16$ for CYTB and COX1, $5.307e-13$ for COX2 and $5.974e-11$ for COX3) (S5 Table).

## In a concordantly evolving site pair, allele arising at one site is dependent on the previously arisen allele at the other site

If concordance of evolution results from epistatic interactions between the leading and trailing sites, the leading substitution might affect not just the overall rate but also relative probabilities of different substitutions at the trailing site. Therefore, we hypothesized that at a trailing site of a concordantly evolving site pair, the probabilities of different amino acid substitutions (from the same ancestral amino acid) will be biased, compared to other substitutions at this site. To measure the dependence of derived alleles at the trailing site on the derived alleles at the leading site, we use the mutual allele preference (MAP) statistic. MAP takes values between 0 and 1; values below 0.5 correspond to independence of derived alleles, and values greater than 0.5 indicate that the derived allele at one site depends on the derived allele at the other site. MAP = 1 corresponds to perfect association between the two alleles, so that all substitutions to allele B at the trailing site are preceded by a substitution to allele A at the leading site, and no alleles other than B arise on the background of allele A (see Methods).

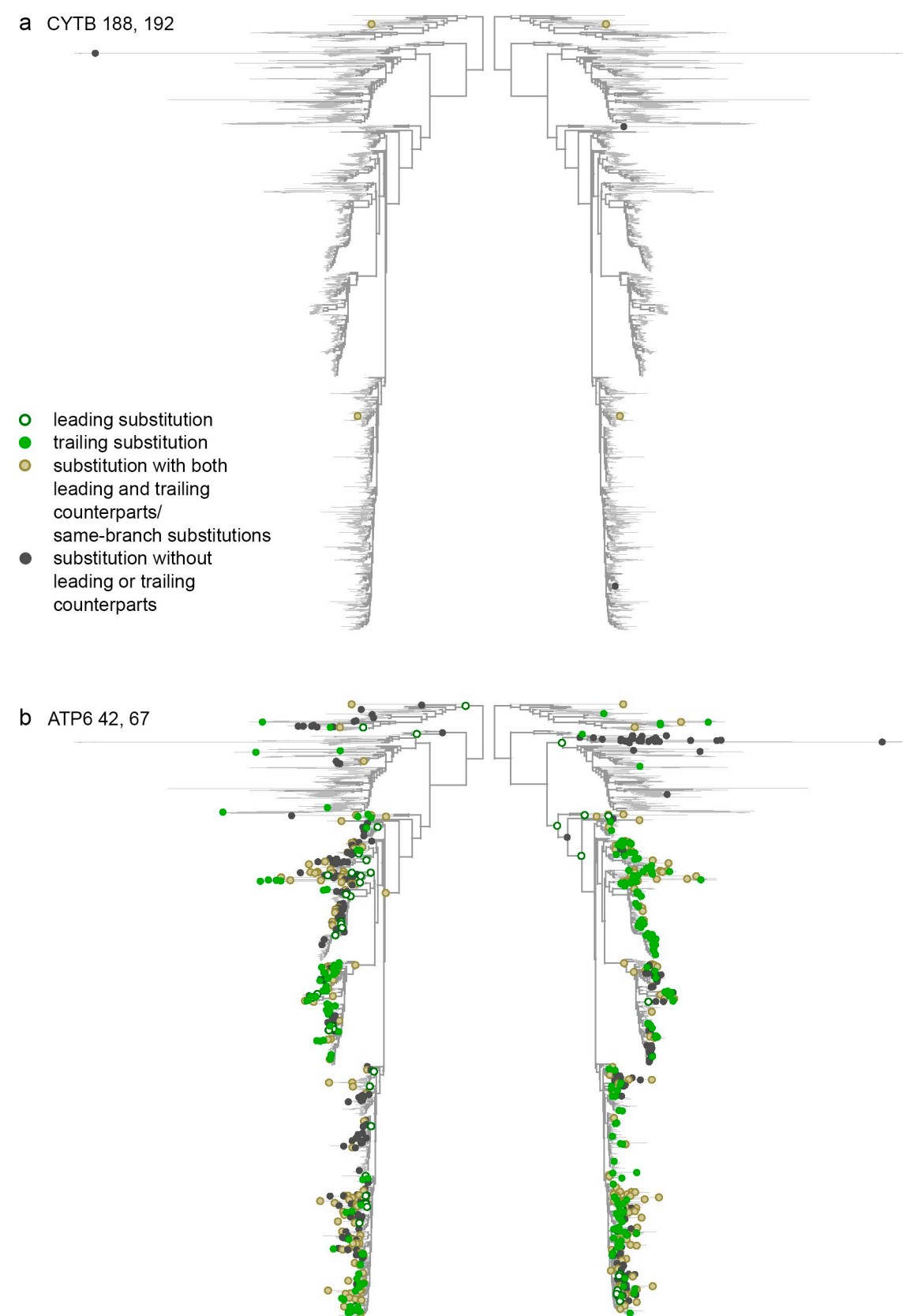

a  CYTB 188, 192

○  leading substitution
●  trailing substitution
◉  substitution with both
   leading and trailing
   counterparts/
   same-branch substitutions
●  substitution without
   leading or trailing
   counterparts

b  ATP6 42, 67

**Fig 4. Concordant evolution of pairs of sites.** Each panel shows the same phylogeny twice, with the dots in each side of the panel indicating substitutions at the corresponding site. a. Concordant evolution of sites 188 and 192 in CYTB is likely explained by epistatic interaction, as revealed by strong clustering of consecutive substitutions in the two sites (all four of which occurred very rapidly one after the other) and lack of clustering of non-consecutive substitutions. b. Concordant evolution of sites 42 and 67 in ATP6 is likely explained by concurrent additive episodic selection, as revealed by strong clustering of both consecutive and non-consecutive substitutions in the same parts of the tree.

Concordantly evolving site pairs have greater MAP than other site pairs for four of the proteins: ATP6, CYTB, COX1 and COX2 (S6 Table); for COX3, no difference was observed. For concordantly evolving site pairs, MAP was also significantly positively correlated with the association statistic (S7 Fig, Spearman's correlation rho between 0.06 and 0.13; S7 Table and S6 Spreadsheet). The bias in the direction of substitutions was particularly pronounced for spatially contacting sites. Indeed, among concordant site pairs, those that were in contact in a structure had higher MAP values than those that were not (S8 Table), and the MAP values were significantly negatively correlated with 3D distances (Spearman's correlation rho between—0.21 and -0.15; S8 Table).

Notably, MAP values were particularly elevated at those concordantly evolving sites in which there was no tendency for nonconsecutive substitutions to occur in proximity to one another at phylogenetic trees. Indeed, the concordantly evolving site pairs with high values of MAP tended to have low absolute values of clustering z-scores (Spearman's correlation rho between -0.25 and -0.10; S9 Table). This indicates that cooccurrence of amino acids is a signal of epistatic, rather than non-epistatic episodic, selection.

## Detecting groups of coevolving sites

Based on the observed pseudo-correlations, we aimed to construct a coevolution graph in which vertices correspond to individual sites, and edges correspond to either positive or negative associations between them (S1 Data). For this, we transformed the matrix of pseudo-correlations by singling out only significant associations, and, among the positively associated site pairs, only those responsible for direct, rather than transient, correlations (see Methods). The resulting association statistics were then used to weight edges of the coevolution graphs (S10 Table). The resulting coevolution graphs were then subdivided into subgraphs corresponding to coevolving groups of sites using modularity method for signed graphs [49]. This resulted in groups of sites such that the density of positive edges was high within groups and low between groups, and the density of negative edges was low within groups and high between groups. Here, the within-group edge density was defined as the ratio of total weight of graph edges connecting vertices in the same group to the total weight of all edges, and the inter-group edge density was defined as the complementary value. For each mitochondrial protein, between 4 and 8 groups of sites were thus defined, together including between 85% and 96% of reference protein lengths (Table 3, Fig 6, S7 Spreadsheet and S1 Data).

Indeed, sites within groups were frequently in contact (Table 3 and Fig 6): for each protein, the density of contacts between sites in coevolving groups was higher than expected (P<1e-4). A significantly elevated number of contacts with sites of the same group was also observed for the majority of individual coevolving groups of sites (Table 3), and these groups include the majority of sites (for the p<0.043 threshold corresponded to Benjamini–Hochberg procedure for 5% FDR correction, 100% for COX2 and CYTB, 96% for COX1, 90% for COX3 and 58% for ATP6).

## Coevolving groups of sites and interfaces of protein-protein interactions

Next, we tested whether the grouping of sites into coevolving sets tends to be non-random with respect to their involvement in inter-protein interactions with other proteins in the

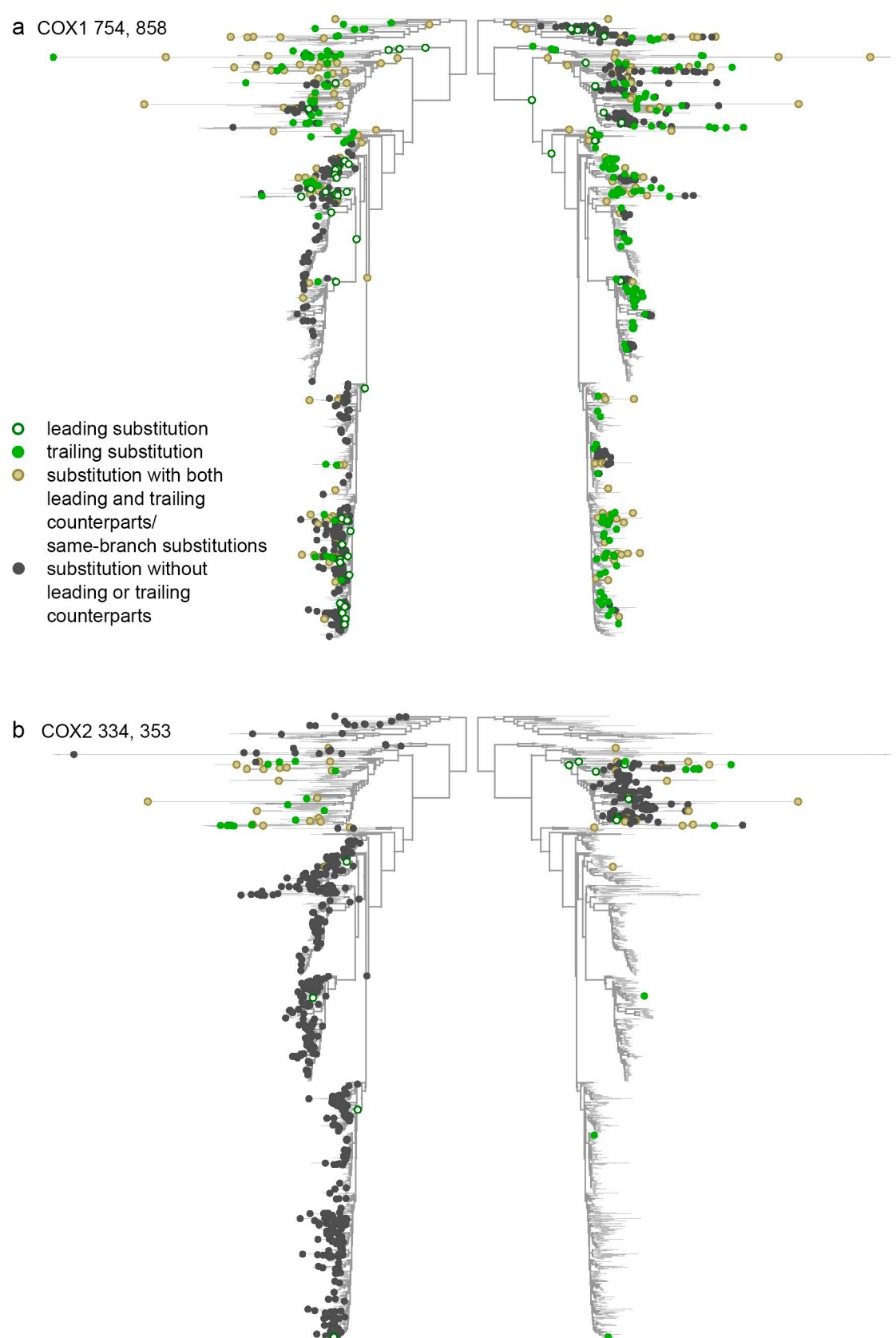

**Fig 5. Discordant evolution of pairs of sites.** a. Discordant evolution of sites 754 and 858 in COX1 is likely explained by epistatic interaction. This is indicated by a deficit of consecutive substitutions together with clustering of non-consecutive substitutions, suggesting negative epistasis that is limited to substitutions at a subset of lineages. b. Discordant evolution of sites 334 and 353 in COX2 is likely explained by additive episodic selection. This is indicated by a deficit of consecutive substitutions together with repulsion of non-consecutive ones, suggesting alternating episodes of accelerated evolution at these two sites.

respiratory complexes (either mitochondrial- or nuclear-encoded). In doing so, we controlled for the previously established fact that the sites in coevolving groups tend to be colocalized within a protein. For COX1, COX2, COX3 and ATP6, although not for CYTB, we found that the coevolving groups of sites were non-random with respect to protein-protein interfaces (Figs 7 and 8 and S11–S15 Tables).

To better understand the link between coevolution and involvement in protein-protein interfaces, we considered each group of concordantly evolving sites in each protein individually. In three of the analyzed proteins, COX1, COX2 and COX3, we found that some groups favored such interfaces, while other groups avoided them (Fig 7 and S11–S13 Tables). In ATP6, most of the sites belonging to the group 3 were located in the helices H5 and H6 and faced the c-ring of the ATP-synthase complex, forming hydrophilic cavities essential for proton transport through the membrane [50]. The conserved arginine 159 crucial for proton translocation [51] also belongs to group 3 (Fig 8).

## Concordant evolution of groups of sites in different OXPHOS proteins

Groups of sites involved in coevolutionary interactions may undergo coordinated acceleration and deceleration of the overall rate of evolution. We aimed to understand when such

**Table 2. Numbers of concordantly and discordantly evolving site pairs mappable to protein structures and correlations between strength of excess (deficit) of rapid consecutive substitutions and distances between sites on protein structures.**

| gene | concordant (+) discordant (-) | nominal p-value threshold for FDR<0.3 | #pairs | rho (Spearman's) association statistics vs. 3D distances | rho p-value association statistics vs. 3D distances |
|------|---|---|---|---|---|
| ATP6 | | | | | |
| | + | 0.0262 | 1764 | -0.15 | 6.96E-10 |
| | - | 0.05 | 5213 | -0.01 | 0.4795 |
| CYTB | | | | | |
| | + | 0.0099 | 2216 | -0.24 | <2.2e-16 |
| | - | 0.05 | 12782 | -0.01 | 0.2759 |
| COX1 | | | | | |
| | + | 0.0316 | 12114 | -0.24 | <2.2e-16 |
| | - | 0.05 | 11431 | -0.004 | 0.63 |
| COX2 | | | | | |
| | + | 0.0135 | 875 | -0.24 | 8.64E-13 |
| | - | 0.05 | 3522 | -0.07 | 6.37E-05 |
| COX3 | | | | | |
| | + | 0.0368 | 3375 | -0.16 | <2.2e-16 |
| | - | 0.05 | 4637 | -0.07 | 6.18E-06 |

For each protein the following statistics are shown: the numbers of significantly concordantly ('+') and discordantly ('-') evolving site pairs with known distances between sites in protein structures, those nominal p-values were below thresholds corresponding to FDR<0.3 (#pairs). For concordantly evolved site pairs the association statistics equal to partial correlations and for discordantly evolved site pairs they equal to pseudo-correlations. The Spearman's correlations (rho) between distances on the protein structures and association statistics as well as corresponding p-values (rho p-value) are shown. For concordantly evolving pairs of sites, a significantly negative value of rho means that strongly associated sites tend to be closer on the structure; for discordantly evolving pairs of sites, a significantly negative rho means that strongly associated sites tend to be apart from each other.

**Table 3. Sites within a coevolving group are colocated on protein structures.**

| gene | group | #sites | Observed, in-group contact density | Expected, in-group contact density | p-value |
|------|-------|--------|-----------------------------------|-----------------------------------|---------|
| atp6 | | | | | |
| | 1 | 64 | 0.37 | 0.33 | 0.0651 |
| | 2 | 66 | 0.5 | 0.34 | <1e-4* |
| | 3 | 46 | 0.54 | 0.23 | <1e-4* |
| | 4 | 17 | 0.09 | 0.08 | 0.3404 |
| | total | 193 | 0.43 | 0.29 | <1e-4† |
| cytb | | | | | |
| | 1 | 84 | 0.32 | 0.23 | <1e-4* |
| | 2 | 78 | 0.27 | 0.21 | 0.0059* |
| | 3 | 64 | 0.5 | 0.17 | <1e-4* |
| | 4 | 49 | 0.2 | 0.13 | 0.0042* |
| | 5 | 39 | 0.31 | 0.1 | <1e-4* |
| | 6 | 23 | 0.17 | 0.06 | 0.0005* |
| | 7 | 15 | 0.26 | 0.04 | <1e-4* |
| | 8 | 10 | 0.43 | 0.03 | <1e-4* |
| | total | 362 | 0.31 | 0.17 | <1e-4† |
| cox1 | | | | | |
| | 1 | 130 | 0.28 | 0.16 | <1e-4* |
| | 2 | 116 | 0.25 | 0.14 | <1e-4* |
| | 3 | 103 | 0.49 | 0.12 | <1e-4* |
| | 4 | 52 | 0.26 | 0.06 | <1e-4* |
| | 5 | 49 | 0.24 | 0.05 | <1e-4* |
| | 6 | 17 | 0.03 | 0.02 | 0.1085 |
| | total | 467 | 0.46 | 0.21 | <1e-4† |
| cox2 | | | | | |
| | 1 | 50 | 0.21 | 0.15 | 0.0037* |
| | 2 | 32 | 0.18 | 0.09 | 0.0005* |
| | 3 | 44 | 0.18 | 0.13 | 0.0056* |
| | 4 | 34 | 0.27 | 0.09 | <1e-4* |
| | 5 | 24 | 0.14 | 0.06 | 0.0021* |
| | 6 | 10 | 0.09 | 0.02 | 0.0039* |
| | total | 194 | 0.32 | 0.19 | <1e-4† |
| cox3 | | | | | |
| | 1 | 74 | 0.25 | 0.19 | 0.0016* |
| | 2 | 64 | 0.2 | 0.16 | 0.0077* |
| | 3 | 46 | 0.25 | 0.11 | <1e-4* |
| | 4 | 26 | 0.13 | 0.06 | 0.0002* |
| | 5 | 22 | 0.07 | 0.05 | 0.1005 |
| | total | 232 | 0.34 | 0.24 | <1e-4† |

For each protein, we partitioned the vertices in the coevolution graph into coevolving groups of sites. Contact graph of a protein represents the physical contacts between sites on protein structures. For each protein, for each coevolving group as well as for the entire graph the mean fraction of edges connecting a site with other sites in the same group, the corresponding expected contact density on random partitions of sites, and the corresponding p-values are shown. We applied the Benjamini–Hochberg correction at the 5% alpha level for tests for individual groups. The p-value threshold corresponding to this correction was 0.043.

*, significant under the Benjamini–Hochberg correction for multiple testing (for groups)

†, significant under the Bonferroni correction for multiple testing (for genes).

Next, we asked whether coevolving groups of sites correspond to clusters in the 3D structure of the protein. For this, for each protein, we constructed a second graph, referred to as a contact graph. In this graph, vertices again correspond to sites, but there is just one type of edge: two sites are connected if the minimal distance between heavy atoms of their correspondent residues is under 4Å. Considering each group of sites in the coevolution graph of each protein, we then asked whether the corresponding subgraph is tightly connected in the contact graph.

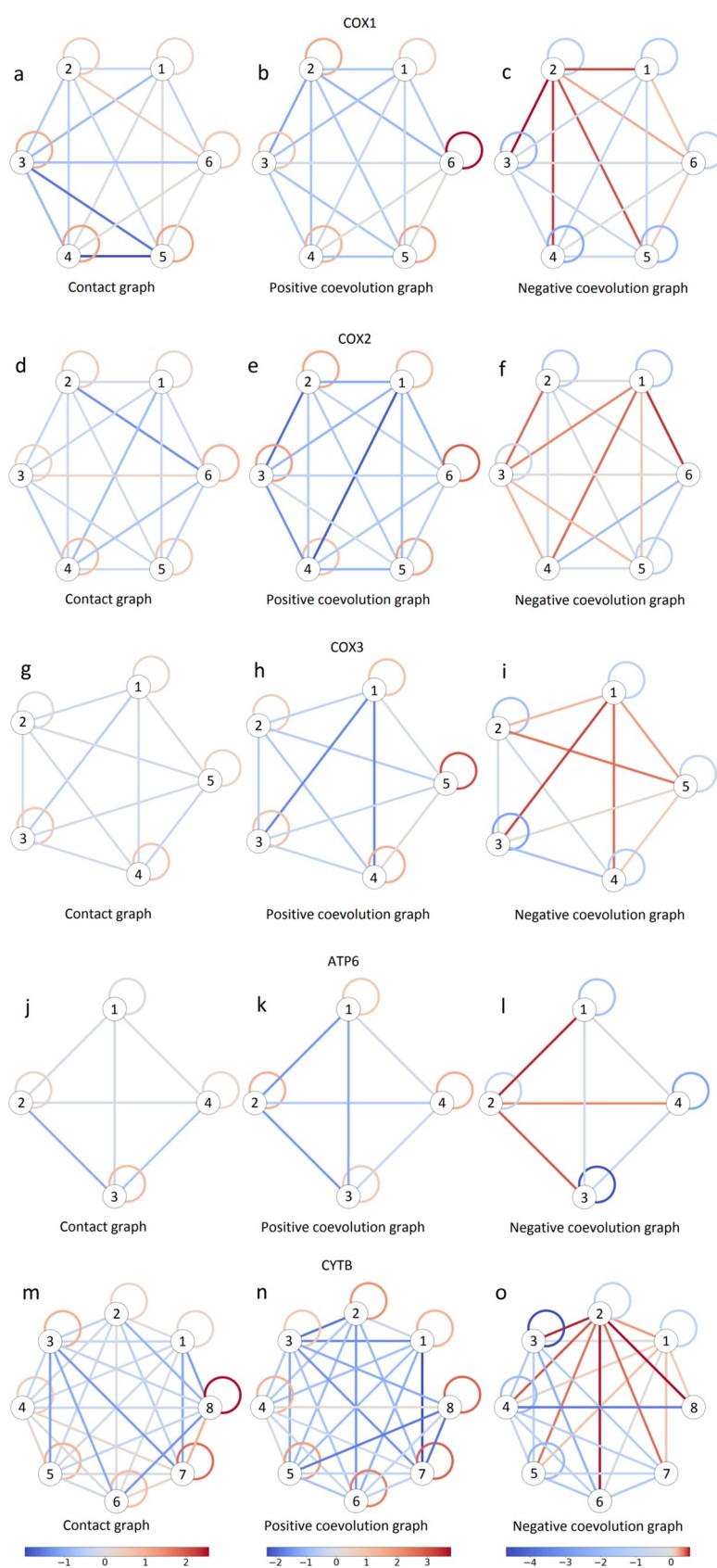

**Fig 6. Schematic representation of contact and coevolution graphs for COX1 (a-c), COX2 (d-f), COX3 (g-i), ATP6 (j-l) and CYTB (m-o) proteins.** Left column (a, d, g, j and m), contact graphs; middle column (b, e, h, k and n), positive edges in coevolution graphs; right column (c, f, i, l and o), negative edges in coevolution graphs. Each group of coevolving sites is represented by a circle. Links connecting circles represent between-group edges, and links connecting a circle to itself represent within-group edges. The color represents the difference of logarithms of the number of edges (left column) or sum of the weights of edges of the corresponding type (center and right columns) and their expected values obtained from a random model used by the vertex clustering algorithm [46]. For contact graphs, groups of coevolving sites are enriched in contacts on the protein structure: the number of edges connecting sites within a group is greater than expected from the random model. For coevolution graphs, the groups of coevolving sites have larger normalized total weights of positive edges within groups than expected from the random model; by contrast, negative edges tend to have greater than expected normalized total weights between groups.

acceleration or deceleration had taken place. For this, for each protein, we identified a number (between 41 and 106) of branches of the phylogenetic tree out of the total of 4349 internal branches where the relative frequencies of substitutions had changed between coevolving groups of sites, so that the clade of the descendants of this branch has a substitution frequency significantly different from that in the rest of the tree (S16 Table).

We asked whether the identity of such branches was concordant between proteins. To test this, we considered the 2200 branches with enough mutations in coevolving groups to test for a change in mutation frequencies in all five proteins. Since it was impossible to unambiguously position such changes when they had occurred in the two consecutive branches (see Methods), for this test, we shifted the inferred position of each change by one branch towards the root of the tree (i.e., to the parental branch). This resulted in 1562 parental branches that could correspond to frequency shifts at one or both of the daughter branches (Table 4 and S1 Data). Depending on the protein, at between 28 and 62 of these branches, such frequency shifts were actually observed (S16 Table).

We compared the numbers of branches such that the specified number of different proteins (between 0 and 5) changed substitution rates concordantly on this branch. The expected values were obtained from a null-model assuming the same number of rate changes occurring in each protein (S16 Table) independently of the other proteins, with the probability of a change in substitution rates on a particular branch to be proportional to its length.

The identity of the branches corresponding to frequency shifts was unexpectedly similar between proteins (Table 4). Six branches were concordantly represented for all five proteins; they corresponded to the last common ancestors (LCA) of Fungi and Metazoa, Protostomia and Deuterostomia, Echinodermata+Hemichordata and Chordata, Actinopterigia and Sarcopterigia+Tetrapoda, Lophotrochozoa and Ecdysozoa, Coleoptera and other Holometabola. Four branches were each represented for four proteins; these were the LCAs of Leotiomyceta and Saccharomycetales, Ensifera and Acrididea, Otomorpha and Euteleosteomorpha, Neuropterida and other taxa within the Endopterygota group. 13 branches, including the LCA of Mammalia and Diapsida, were each observed in three proteins; and 24 branches were observed in two proteins.

## Discussion

Epistatic interactions leave a footprint in the evolutionary history of a protein. Explicit reconstruction of past evolution of individual sites allows inferring pairs of sites such that substitutions at them are correlated in time, a pattern which may arise due to positive epistasis. This idea has been used in an approach designed to infer as interacting site pairs those where substitutions occur unexpectedly rapidly after one another [44,45,52,53]. This, however, has only allowed detecting positive epistasis, i.e., the situation in which the first of the two substitutions in a pair increases the selective advantage of the second substitution.

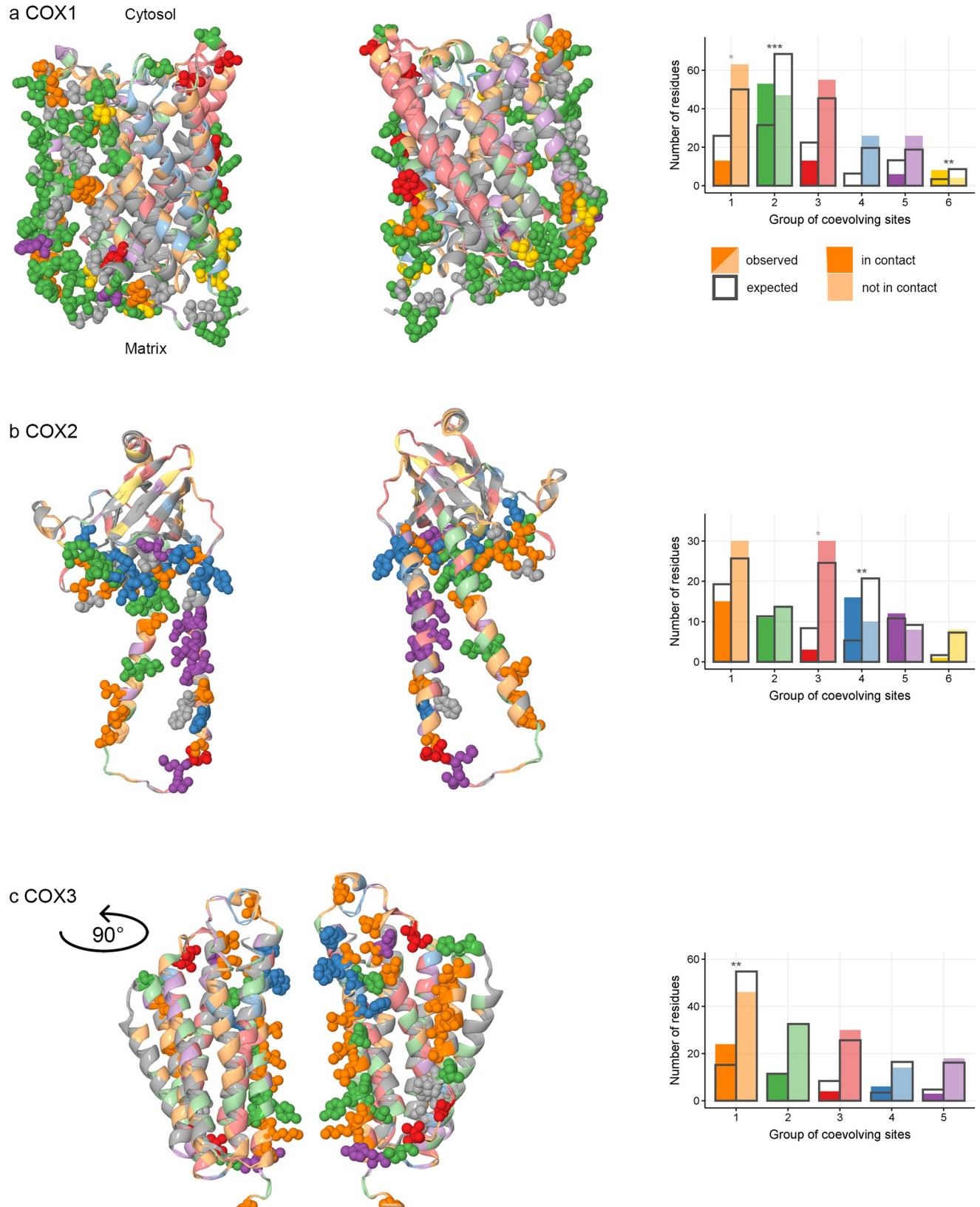

**Fig 7. Groups of concordantly evolving sites and interactions between subunits of COX.** For each mitochondrial-encoded COX protein: COX1 (a), COX2 (b) and COX3 (c), structures of two identical subunits within the crystallized homodimer are shown (PDBID: 1occ). The groups of coevolving sites are color-coded. (a,b,c on the left) protein structure of COX. Residues at sites involved in interactions with other COX proteins are shown as spheres; residues at sites on protein surfaces not involved in interprotein interactions, as ribbons; the remaining sites (internal sites or sites that were excluded from analysis) were colored in gray. (a,b,c on the right) the numbers of sites which are in contact and numbers of sites which are not in contact with other COX proteins in the protein structure, compared to the expected values. Significant differences are marked with asterisks (*, p<0.025; **, p<0.005; ***, p<0.0005). For COX1 (a), interactions with nuclearly encoded COX proteins are considered; for COX2 and COX3 (b, c), interactions with other mitochondrial-encoded COX proteins are considered.

Most existing methods for detection of interactions between sites, such as DCA-based methods, use multiple sequence alignments without explicitly accounting for evolutionary relationships between considered species. Accounting for the phylogeny provides several advantages. First, the MSA-based methods implicitly assume independence of lineages, considering differences in evolutionary distances between lineages as a nuisance factor. By contrast, phylogeny-based methods provide a formal way to account for non-independence between evolving lineages. Second, rooted phylogenies provide explicit polarization of trait states, allowing to distinguish between ancestral and descendant states. In turn, this allows to detect not only positive, but also negative associations between allele pairs.

Here, we make use of this latter advantage. We extend our phylogenetic approach to detect the second possible type of epistatic interactions: negative epistasis. In a pair of negatively epistatically interacting sites, the first substitution reduced the fitness benefit conferred by the second substitution, making the second substitution less probable. As a result, substitutions at

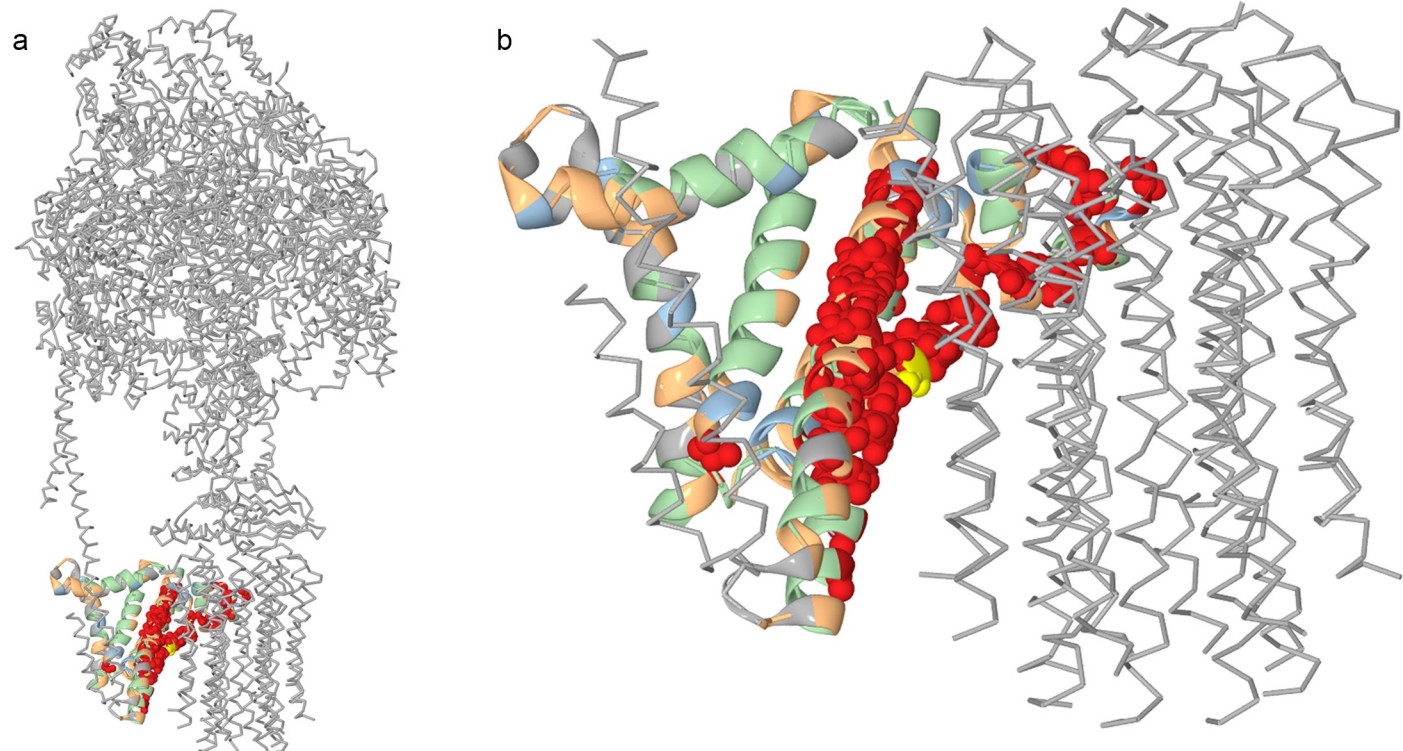

**Fig 8. Groups of concordantly evolving sites and function of ATP6 in ATP synthase.** For the mitochondrial-encoded ATP6 protein, the groups of coevolving sites are color-coded. Residues at sites of group 3 are shown as spheres; residues at other sites, as ribbons. Most residues at group 3 sites face the rotor part of the ATP synthase and may contribute to proton transport across the membrane. Yellow, Arg159 which also belongs to group 3.

**Table 4. Substitutions rates in coevolving groups of sites have changed concordantly in the evolution of Metazoa and Fungi.**

| No. of proteins that have changed their substitution rates concordantly | 5 | 4 or more | 3 or more | 2 or more | 1 or more | 0 or more |
|---|---|---|---|---|---|---|
| Observed no. of branches with this no. of changes | 6 | 10 | 23 | 47 | 131 | 1562 |
| Expected no. of branches with this no. of changes | 0.2 | 1.0 | 5.0 | 27.8 | 183.1 | 1562 |
| p-value | <1e-4 | <1e-4 | <1e-4 | <1e-4 | 1 | 1 |

negatively epistatically interacting pairs of sites will be "repelled" from one another, leading to a deficit of substitutions occurring one after another at the same lineage.

Using simulations, we show that no signal of concordant or discordant evolution arises when the amino acid propensities at each site are either constant or fluctuating but independent between sites, both when sites are independent or linked. This is as expected: our statistics assess the presence of significant associations between sites, and no such associations can arise in simulations where no interactions between sites are present. Our power to detect positive and negative epistasis differs: positive epistasis is more easily detectable, probably because an excess of rare events (pair of consecutive substitutions) is more evident than their deficit.

We apply our method to the mitochondrial-encoded subunits of the OXPHOS protein complexes. Sites of OXPHOS proteins are known to have undergone changes both in their substitution rates ('heterotachy') and spectra ('heteropecilly') between lineages. Here, we show that much of this change is correlated between sites, either positively or negatively. Positively interacting sites are positioned close to each other in the protein structure, providing an independent validation for our approach.

The substitution rate is affected by mutation rate and selection. While the mitochondrial substitution rates have changed due to changes in the mutation rates [25,54], it is unlikely that such changes are site-specific. By contrast, changes in substitution rate may occur naturally at individual sites due to changes in site-specific amino acid propensities. The observed within-site heterogeneity may be most simply explained by such changes in selection with time.

Fundamentally, concordant evolution of a pair of sites may have two causes. In a static fitness landscape, it can only result from a direct causal link between the substitutions at the leading and the trailing sites, i.e., positive epistasis. In addition, if the fitness landscape is allowed to change, it can be caused by a concurrent change in selection affecting these two sites in some lineages, leading to their episodic evolution. Reciprocally, discordant evolution of a pair of sites may result from negative epistatic interactions between substitutions in them, or from alternating episodes of selection. Clustering of nonconsecutive substitutions can only result from non-epistatic episodic selection, and our observation of such clustering indicates that episodic evolution certainly plays a role. We also provide evidence that the observed patterns are hard to explain without assuming some causal link between the substitutions in the concordantly evolving site pairs, indicating that both episodic and epistatic selection are acting.

By applying community detection methods, we show that the concordantly evolving sites are grouped into domains, with negative epistatic interactions distinguishing these groups from each other. We find that the sites that have changed concordantly are those that are functionally and structurally linked. This is consistent with previous findings obtained by analyzing sequence alignments [9,10,55] and is in agreement with results of theoretical studies which show that structural properties of networks of tightly interacting protein sites depend on the number of acting selective forces [19,21] and the evolutionary history [19]. Such concordance may also be caused by direct pairwise interactions between sites. Even in the absence of direct interactions, groups of coevolving sites may arise [21] due to protein-level selection forces mediated by one-dimensional, or global, epistasis [56,57]. One possible evolutionary constraint shaping the evolution of COX is the maintenance of interactions between proteins

within this complex [58]. Indeed, some of the inferred coevolving groups of sites in COX1, COX2 and COX3 are associated with interactions with other COX proteins, either mitochondrial- or nuclear-encoded or both. Earlier, an excess of biochemically radical substitutions has been observed at interfaces between nuclear and mitochondrial-encoded subunits of COX, suggestive of adaptation [59].

Incompatibilities between nuclear and mitochondrial genomes may play a role in speciation, affecting substitution patterns [22,58,60] in particular, selection for reproductive isolation may cause bursts of substitutions in interfaces between subunits encoded in nuclear and mitochondrial genomes [41,61,62]. The observed coevolution within interface sites may be partially explained by such selective pressures.

To understand what causes changes in substitution rates in groups of concordantly evolving sites, we hypothesized that the evolution of different mitochondrial proteins was also concordant in that evolutionary rate changes were coordinated between different proteins, in addition to their coordination within proteins. Consistently with this hypothesis, we find that different proteins that are subunits of the same as well as different complexes of OXPHOS change substitution rates in groups of sites on coincident branches of the phylogeny. This concordance may reflect changes in the selection pressure on the respiratory function in the process of adaptation to certain ecological niches affecting multiple OXPHOS proteins simultaneously [63]. The branches that had experienced such concordant changes tended to be deeply rooted in the phylogeny, likely indicative of adaptation at the origin of large taxonomic groups. For example, the LCA of mammals indicated concordant changes in substitution spectra for three genes: CYTB, COX3 and ATP6.

Negative epistasis between deleterious mutations has been described from population genomics data [64] and has been postulated to play a major role in maintenance of sexual reproduction [65]. Furthermore, negative epistasis has been observed in experimental evolution. One characteristic pattern is the presence of distinct "seeding" mutations early in the adaptation process that each trigger its own cascade of subsequent adaptive mutations, directing subsequent evolution. The seeding mutations themselves are in negative epistasis, making them effectively mutually exclusive, and leading to substantial randomness in the choice of the particular adaptive path taken by the population [66]. This pattern is indeed theoretically expected both within and between proteins on fitness landscapes with high local ruggedness [67–69] and has been observed in multiple evolutionary experiments [66,68].

A high prevalence of negative epistasis in mitochondrial proteins may have to do with their strong modularity. Conceivably, changes in one domain may trigger subsequent changes in the same domain while increasing the cost of changes in other domains, in line with the "seeding mutations" model [66]. As the respiratory function is carried out by several protein complexes, each consisting of multiple subunits encoded in two genomes with significantly different mutation rates, the negative epistasis between sites or domains of one protein may be driven by the need to support the integrity of this complex system. Our finding that positive epistatic interactions tend to be short-range, and negative, long-range in protein structures (Tables 2 and S3), is consistent with the results obtained in a high throughput mutagenesis experiment in GB1 protein [70].

Whereas bursts of substitutions in correlated sites caused by positive epistasis have been reported previously [44,45,71,72], to our knowledge, negative epistasis has been rarely reported from similar data [40,73]. Families of mitochondrial proteins are an excellent model for the study of negative epistasis between sites because of a high number of substitutions in each site and multiple constraints on their evolution.

## Methods

### Data

Amino acid sequences of five OXPHOS proteins (COX1, COX2, COX3, ATP6 and CYTB) encoded in mitochondrial genomes of 4350 species of metazoans and fungi were obtained from [28]. Each protein was aligned with MAFFT v6.864b [74] using the einsi option. For phylogenetic reconstruction, alignment columns with more than 1% of gaps were excluded, and sequences of the five genes were concatenated. The phylogenetic tree was reconstructed with RAxML 8.0.0 [75] using ITOL taxonomy-constrained topology as described in [28]. Bootstrap support for each branch was obtained using the rapid bootstrap option of RAxML8.0.0 [76]. For ancestral state reconstruction, we excluded columns with ≥10% of internal gaps. Ancestral states were reconstructed with MEGA-CC [77] using "mtREV with Freqs. (+F)" model and Gamma distributed evolutionary rates between sites with 4 discrete Gamma categories. As the length of COX1 exceeds the limit of MEGA, ancestral states were reconstructed separately for two halves of its alignment. For each gene, we excluded sites with less than two substitutions on the tree, thus the numbers of analyzed sites of corresponding multiple alignments were 221 for ATP6, 386 for CYTB, 485 for COX1, 220 for COX2 and 250 for COX3.

3D structures were obtained from PDB (1occ for COX1-3, 5ara for ATP6 and 1bgy for CYTB) [78–80]. Due to all these structures were obtained for bovine proteins, to find the correspondence between columns in MSA and position of sites in protein structures, for each protein we performed a pairwise alignment of *Bos taurus* (TaxID = 9913) protein sequence from our MSA to that of the corresponding protein chain in the PDB file using BlastP [81,82].

### Inference of epistatic site pairs

To detect epistasis between protein sites, we reimplemented the phylogenetic method from [44], with some modifications, using the BioPhylo package for Perl [83]; see S1 Methods for details. For each protein we considered all possible $N^*(N-1)$ unordered pairs of sites, where N is a number of analyzed sites of corresponding multiple alignment. As in [44], for each pair of sites, we calculated the epistatic statistic as the number of pairs of single amino acid substitutions that were consecutive, i.e., fell onto the same phylogenetic lineage and were not separated in the lineage by other substitutions at these two sites. Mutations that followed one another rapidly had higher weight, with exponential penalties for the waiting time of a second mutation in a pair [44]. As in [44], we compared the observed values of the epistatic statistics with those expected if mutations at different sites were distributed independently of each other, preserving the numbers of mutations for each site and for each branch. To generate these null distributions, we used the BiRewire package for R [84]. A total of 10000 sets of mutations were generated in parallel using the GNU Parallel [85] utility. The upper and lower p-values for the epistatic statistic were defined as the percentiles of the null distribution corresponding to the observed values of this statistic.

For each p-value, we estimated the false discovery (FDR) [86,87] rates following the procedure from [44]. Briefly, for 400 random sets of mutations on the phylogeny, we inferred positively and negatively coevolving site pairs. We estimated the FDR as the ratio of the average number of findings (coevolving site pairs with the same or better p-value) over these 400 random sets to the number of findings in the real data.

To make sure that the observed associations between evolutionary processes at different sites were not artifacts of clustering of spurious substitutions in clades with incorrectly reconstructed topologies [88], we performed a separate analysis accounting for the uncertainty in phylogenetic reconstruction as follows. We defined a subset of well resolved branches of the

phylogeny as those with rapid bootstrap [76] support exceeding 95%. These branches split the tree into subtrees with poorly resolved branches. We assumed that the phylogenetic position of the substitutions at well resolved branches was unambiguous. By contrast, the precise number and phylogenetic position of substitutions falling onto a poorly resolved subtree was unknown. We conservatively assumed that each poorly resolved subtree had experienced no more than one substitution at a site. If multiple substitutions within a poorly resolved subtree were reconstructed, we therefore assumed that all but one of these substitutions were spurious. Under this assumption, the phylogenetic position of the only real substitution was not known exactly. We therefore calculated the epistatic statistic as the weighted sum over all of the $n$ potential (reconstructed) positions of this substitution within the subtree, each with the weight of $1/n$.

### Association statistics

For each unordered pair of sites, we defined the pseudo-correlation as z-scores of the sum of the epistatic statistics for the two corresponding ordered pairs, normalized so that the highest value was 1 if positive, or lowest -1 if negative. Next, we aimed to single out the site pairs driving the observed positive pseudo-correlations, and to get rid of spurious positive pseudo-correlations resulting from indirect interactions between sites. For this, following previous studies [7,89,90], for each site pair, we defined the association statistic as follows. If the pseudo-correlation was positive, the association statistic was assumed to equal the corresponding partial correlation calculated by cor2pcor R package (http://www.strimmerlab.org/software/corpcor/) with the correlation shrinkage intensity lambda set to 0.9 [91]; if the pseudo-correlation was negative, the association statistic was assumed to equal the pseudo-correlation itself.

### Simulation of episodic positive selection, clonal interference and hitchhiking

To simulate independent episodic positive selection at individual sites, we used the SELVA phylogenetic simulator [46]. We modelled the evolution of 500 amino acid sites along the reconstructed phylogeny, with individual independently changing fitness vectors which described relative fitnesses of all 20 amino acids at a given site. Initial fitness vectors were sampled from lognormal distribution with mu = 0 and sigma = 0.25; each subsequent vector was sampled from the same distribution independently of the previous one. Events of landscape changes were modelled as a poisson process on the phylogenetic tree with predefined branch lengths, with landscape for one site changing independently on different branches. We chose the parameter of the poisson process to be 0.5 in tree length units, which resulted in an average of 20 landscape changes per site.

To model linked evolution of multiple sites, we used SantaSim, a forward-time simulator of molecular evolution in a population [47]. The simulation started from the population of identical genotypes each of 566 site lengths, the starting genotype was picked randomly. The population size was selected to be constant and equals to 10000 entities. The population was allowed to evolve during 5000 generations with mutation rate 2e-5 mutations per nucleotide site per generation. At all sites, a single amino acid was preferred, with fitness of each of the other amino acids being 0.9 of the best fitness. Every 5 generations, 2 genotypes were randomly picked from the general population for further analysis. The resulting 2000 sequences were used for phylogenetic reconstruction, and the signal of concordant or discordant evolution was measured as described previously.

## Simulation of positive and negative epistasis

To model evolution under epistasis, we used genome-wide forward simulator MimicrEE2 [48]. It allows modelling of epistatic interaction between a pair of loci, directly assigning fitnesses to all possible combinations of binary variants at these loci (*aa*, *aA*, *Aa* and *AA*). To get the fitness of a genome, MimicrEE2 multiplies fitnesses of single variants and fitnesses of variant combinations for specified pairs. Initial population consisted of identical genotypes with allele *a* at each site, with a total of 100 sites: 20 sites (or 10 pairs of sites) in positive epistasis, 20 sites (or 10 pairs of sites) in negative epistasis and 60 neutrally evolving sites with no epistatic interactions. At neutrally evolving sites, all variants had fitness equal to 1. To model positive epistasis between a pair of sites, we assigned fitness 1 to variant combinations *aa* and *AA* and fitness 0.9945 to *aA* and *Aa*, so that the first mutation at one of the sites was deleterious, and consequent mutation at the second site restored the initial fitness. To model negative epistasis, we assigned fitness 1 to combinations *aa*, *aA* and *Aa* and 0.8 to *AA*, so that the first mutation at one of the sites was neutral, and consequent mutation at the second site was deleterious.

We simulated the evolution of haploid population of size 50000 during 5000 generations, with mutation rate 5e-4 mutations per site per generation. Each 250 generations, 50 genotypes were sampled from the population, resulting in 1000 sequences which were further analysed. We reconstructed the phylogeny and then measured the signal of concordant or discordant evolution as described previously.

## Analysis of attraction between nonconsecutive substitutions

For each unordered pair of sites, we measured phylogenetic distances between all pairs of nonconsecutive amino acid substitutions, i.e. substitutions that fell onto different phylogenetic lineages or fell onto the same lineage but were separated from each other by another substitution at one of these sites. To obtain clustering z-scores and p-values, we compared the observed mean values of distances for individual site pairs with those expected under null model (see "Inference of epistatic site pairs"). In the null model, mutations at different sites were distributed independently of each other, preserving the numbers of mutations for each site and for each branch. A total of 200 sets of mutations were generated. Clustering z-scores were defined as z-scores with inverted signs (expected minus observed values of statistic normalized on the standard deviation), thus the statistics were positive if substitutions have been unexpectedly close to each other and negative otherwise. We compared the measure of clustering of nonconsecutive substitutions with the pseudo-correlation that is the measure of excess or deficit of rapid consecutive ones that had positive (negative) values for concordantly (discordantly) evolved site pairs.

## Mutual allele preference statistic

Our method for looking for concordantly and discordantly evolving site pairs ignores identities of amino acids in substitutions at protein sites. In reality, for a pair of epistatically interacting sites, a strong dependence between alleles in these sites is also expected because fitness of an amino acid in one site is dependent on which amino acid occupies the other interacting site. We measure this dependence using the mutual allele preference statistic (MAP). To define MAP, for the leading site s1, we denote leading substitutions from any ancestral allele a into a specific derived allele A as (s1•A), and into any other derived allele as (s1•!A). For the trailing site s2, we denote trailing substitutions from the ancestral allele b in a lineage descendant from any leading substitution at s1 as (s1,b•s2). Finally, we denote the trailing substitutions from b at s2 in a lineage descendant from any leading substitution at s1 as (s1,b•s2•B) and (s1,b•s2•!B)

depending on whether they result in a specific derived allele B or in any allele other than B respectively.

Using this notation, we define the MAP statistic for the derived allele B in the site s2 in the context of the derived allele A in the site s1 as follows: M(1,2,A,B,b) = P(s1•A)P(B|s1•A,b•s2) +P(s1•!A)P(!B|s1•!A,b•s2), where P(s1•A) = n(s1•A)/n(s1) is the probability of substitution from any allele in the site s1 into the allele A, P(B|s1•A,b•s2) = n(s1•A,b•s2•B)/ n(s1•A,b•s2) and P(!B|s1•!A,b•s2) = 1-(n(s1,b•s2•B)-n(s1•A,b•s2•B))/(n(s1,b•s2)-n (s1•A,b•s2)).

MAP is defined only if the substitutions in s2 from the ancestral allele b occur against the background of at least two different alleles at s1 and result in at least two different derived alleles. Thus, formally the following constraints should hold: n(s1•A,b•s2)>0, n(s1•!A,b•s2) = n(s1,b•s2)-n(s1•A,b•s2)>0, n(s1,b•s2•B)>0 and n(s1,b•s2•!B) = n(s1,b•s2)-n(s1,b•s2•B)>0. For each site pair, we averaged the values of MAP across all combinations of alleles (A,b,B) for which the MAP statistic was defined.

## Construction of coevolution graphs

For each protein, we then used the values of the association statistic to construct the coevolution graph as follows. All variable sites were represented as graph nodes. We connected a pair of nodes with a "positive" edge if the corresponding site pair had a positive association statistic or with a "negative" edge if the corresponding site pair had a negative association statistic and upper p-value for the positive edge or lower p-value for the negative edge were below the minimum of two values: 0.05 and a p-value threshold corresponding to FDR<0.3.

To identify groups of coevolving sites, we then applied a vertex clustering algorithm optimizing graph modularity [49] implemented in the louvain package (https://pypi.org/project/louvain/).

## Overlaying coevolution and contact graphs

If the inferred coevolution graphs reflect the structural constraints on protein evolution, sites corresponding to adjacent vertices in that graph can be expected to be in spatial contacts. To test this, we constructed a contact graph with vertices representing sites, and edges corresponding to contacts in the protein structure. Following earlier studies, we defined a site pair to be in contact if the minimal distance between the heavy atoms of their residues was <4 angstroms [32].

For each group of coevolving sites, we identified the subgraph of the contact graph corresponding to these sites, and defined the contact density statistic as follows. For each group, we calculated the ratio of the number of edges connecting vertices within the group to the total number of edges which had at least one vertex in this group. For the entire protein, we calculated the ratio of the number of edges having both vertices in the same group to the total number of edges in the contact graph.

## Associations between groups of coevolving sites and protein-protein interface sites

We estimated the associations between coevolving groups and protein-protein interaction interfaces, defined as follows. Following Aledo et. al. [32,92], we classified the amino acid residues with solvent accessibilities in isolated subunit below 5% as buried; those sites were excluded from the contact graph and not considered further in this analysis. The remaining exposed residues were partitioned into contact residues that had contacts with other subunits in the complex; exposed noncontact interface (ENC_interface) residues that had solvent accessibility within the complex lower than that as an isolated subunit; and the remaining exposed

noncontact noninterface residues that were on the protein surface but not involved in interactions with other subunits. Sites in MSA were classified as the corresponding residues of the *Bos taurus* protein. We separately estimated associations of coevolving groups of sites with contact sites and with interface sites, a larger set defined as the union of contact and ENC_interface sites.

The vertex clustering algorithm partitioned protein sites into groups of coevolving sites. We asked whether these groups were enriched or depleted in contact or interface sites, referred to as the testing subsets. To test the null hypothesis of independence, we constructed the contingency table and calculated the chi square statistic. Additionally, for each group of coevolving sites, we estimated the Jaccard index, i.e., the number of vertices common to the testing subset and the considered group of coevolving sites, divided by the number of vertices in either of these sets.

Groups of coevolving sites as well as many of the testing subsets formed dense clusters in the protein structure. We were concerned that significant associations between these characteristics could spuriously arise due to such spatial clustering rather than due to interactions between sites. To control for this, we compared the observed values of the statistics with those expected from random groups of sites with the same extent of clustering in the spatial structure as the testing subset. For this, we sampled random subgraphs of the contact graph that had the same number of vertices and equal or greater number of edges connecting them as the testing subset, and used these samples to estimate the expected counts for the contingency tables and to obtain the p-values. To perform this sampling, we implemented an algorithm similar to the algorithm of uniform sampling of connected subgraphs with predefined numbers of vertices [82], with two differences. First, our method rejected subgraphs having fewer edges than the subgraph of the testing subset. Second, we allowed disconnected subgraphs as follows. If the testing subset corresponded to a disconnected subgraph, we performed sampling for each connected component separately, but prohibited the sampled subgraphs corresponding to different components from containing overlapping sets of vertices. For this, we randomly ordered the connected components; sampled a random connected subgraph corresponding to the first component; removed its vertices from the graph; and repeated the procedure for all subsequent components. Sometimes it was impossible to sample a connected subgraph with the given number of edges from the remainder of the graph. To address this, we limited the number of sampling trials to 10000; if no trial succeeded in finding a suitable subgraph, we rolled the algorithm one step back, sampling a different random subgraph for the previous component. Each subgraph thus sampled defined a binary partition of sites.

## Inference of episodic evolution

We analyzed the distribution of substitutions over the phylogeny, identifying the phylogenetic branches corresponding to changes in substitution accumulation rates in some groups of coevolving sites. For this, we used the following procedure. For each branch of the tree i, we calculated the vector $n_i$ of the numbers of substitutions that occurred at each group of coevolving sites on this branch, and $m_i$ as the sum of $n_k$ across all branches k descendant to i. We then traversed the tree, looking for branches corresponding to significant changes in this vector. First, we defined the root branch as the "current ancestor". Next, starting from it, we traversed the tree towards the terminal branches. For each branch i encountered in this process, we compared the vector of substitutions that occurred on this and all subsequent branches $v_i = n_i+m_i$ to the vector of substitutions that occurred in all the remaining branches descendant to the current ancestor $c_i = m_{a(i)}-v_i$, where $a(i)$ is the current ancestor to i. The two vectors were compared using the Fisher's exact or chi squared tests as implemented in the fisher.test package of

R, with the Bonferroni correction for the number of internal nodes tested. In each comparison, the groups that had no substitutions in the subtree of the current ancestor were excluded. If $v_i$ and $c_i$ were significantly different, we assumed that the branch i corresponded to a significant change in the relative substitution frequencies between groups. In this case, we redefined the current ancestor as i, and repeated the procedure for descendant branches. If the total number of substitutions in a subtree of a branch was very low (equal to the number of groups or less), the test was not further applied to descendant branches.

## Estimation of concordance of episodic evolution between OXPHOS proteins

We asked whether the identified phylogenetic branches corresponding to changes in relative substitution frequencies between groups of coevolving sites were coincident between different OXPHOS proteins. For this, first, we obtained the set of branches that were tested for the potential changes in relative substitution rates in all genes, excluding those branches that were not tested in some of the genes because there were not enough substitutions (see above). For each gene, we also identified the subsets of branches corresponding to changes in relative substitution frequencies. In cases when the current ancestor was the immediate ancestor of the tested branches, it was impossible to decide which of the two sister lineages (or both of them) actually experienced the change. Therefore, we positioned the change with the precision of up to two sister branches. Each such pair could be uniquely identified by the name of the parental branch.

We tested the significance of the overlap between the branches corresponding to changes in relative substitution frequencies between different genes using a permutation test, assuming that a longer branch was more likely to experience a significant change than a short branch. For this, we calculated the numbers of branches corresponding to changes in zero, one, two, etc., five genes. To calculate the expectation for this value, we generated 10000 permutations, randomly picking for each gene the same number of branches as in the data, each with the probability proportional to the sum of the lengths of its two daughter branches. Finally, we calculated the probabilities to observe the specified number of concordant events for k or more genes, where k = 0, 1, 2,..., 5.

## Quantification of data underlying plots and graphs

We quantified the data underlying the plots and graphs resulting from our analyses in S1–S7 Spreadsheets.

## Supporting information

**S1 Methods. Inference of epistatic site pairs.** Identifying interactions between sites. Visualizing contact and coevolution graphs.
(PDF)

**S1 Table. Numbers of concordantly (+) and discordantly (-) evolved site pairs predicted by the method with applied correction for phylogenetic uncertainty.** For each gene, the number of predicted site pairs (#pairs) and the nominal p-values corresponding to FDR<0.3 are shown.
(DOCX)

**S2 Table. Correlations between strengths of clustering of non-consecutive substitutions (clustering z-scores) and the values of excess or deficit of rapid consecutive substitutions (association statistics).** For concordantly evolving pairs, positive correlation of association

statistics with clustering z-scores means that the excess of rapid consecutive substitutions is accompanied by more prominent "clustering" of non-consecutive substitutions. Similarly, for discordantly evolving pairs, positive correlation means that the deficit of rapid consecutive substitutions is accompanied by remoteness of non-consecutive ones. Thus, in both cases, positive correlation implies that episodic selection contributes to the association statistics.
(DOCX)

**S3 Table. Numbers of concordantly and discordantly evolving site pairs mappable to the protein structures, and correlations between strength of excess (deficit) of rapid consecutive substitutions and distances between sites on protein structures.** For each protein the following statistics are shown: the numbers of significantly concordantly ('+') and discordantly ('-') evolving site pairs with known distances between sites in protein structures, those nominal p-values were below thresholds corresponding to FDR<0.3 (#pairs). The strength of excess (deficit) of consecutive substitutions is measured by two statistics—the pseudo-correlation and partial correlations (type of statistics). For concordantly evolved site pairs the association statistics equal to partial correlations and for discordantly evolved site pairs they equal to pseudo-correlations. The Spearman's correlations (rho) between distances on the protein structures and association statistics as well as corresponding p-values (rho p-value) are shown. The Spearman's rho between 3D distances and pseudo-correlations for concordantly evolved are also provided. For concordantly evolving pairs of sites, a significantly negative value of rho means that strongly associated sites tend to be closer on the structure; for discordantly evolving pairs of sites, a significantly negative rho means that strongly associated sites tend to be apart from each other. Some data from this table are presented in Table 3 in the main text of the manuscript.
(DOCX)

**S4 Table. Comparison of distances on protein structures for concordantly and discordantly evolved site pairs.**
(DOCX)

**S5 Table. Numbers of contacting pairs among concordantly and discordantly evolving site pairs.** The concordantly evolving site pairs are enriched by pairs of sites that are in contact on protein structures, oppositely, discordantly evolving site pairs are depleted by contacting site pairs. For each protein the following statistics are shown: the numbers of significantly concordantly ('+') and discordantly ('-') evolving site pairs with known distances between sites in protein structures, those nominal p-values were below thresholds corresponding to FDR<0.3 (#pairs), the number of contacting site pairs among the predicted concordantly and discordantly evolved pairs (#contacts) and proportions of total numbers of contacting pairs of site to total numbers of analysed site pairs (#all contacting pairs/#all pairs). Two sites are considered to be in contact on a corresponding protein structure if the minimal distance between heavy atoms of their side residuals is below 4A threshold.
(DOCX)

**S6 Table. Mutual allele preference statistics (MAP) for concordantly evolving site pairs and other site pairs.** Higher values of MAP correspond to stronger dependencies of amino acid substitution probabilities in one site on the background amino acid in another site in a pair.
(DOCX)

**S7 Table. Correlations of the mutual allele preference statistic (MAP) and association statistics for concordantly evolving site pairs.**
(DOCX)

**S8 Table. Comparison of the mutual allele preference statistic (MAP) for proximal and distant concordantly evolving site pairs.** Among concordantly evolving site pairs the pairs of sites proximal on protein structures also have higher values of MAP statistic than pairs of distant sites. Mean values of MAP for contacting and non-contacting on protein structures concordantly evolved site pairs their standard deviations and p-values of Mann Whitney U-test are shown. The Spearman's correlation (rho) of MAPs and distances on protein structures and probabilities that observed correlations equal to zero (P-val., rho) are also provided.
(DOCX)

**S9 Table. Correlation between absolute values of clustering z-score and the mutual allele preference statistic (MAP) for significantly concordantly coevolving site pairs.** Low absolute values of the substitution clustering z-score statistic correspond to high values of mutual allele preference. The Spearman's correlation (rho) of MAPs and absolute values of substitution clustering z-scores and probabilities that observed correlations equal to zero (P-val., rho) are shown.
(DOCX)

**S10 Table. Basic characteristics of coevolution graphs.** For each protein the number of analysed sites of a corresponding multiple alignment (#sites), the number of vertices in a coevolution graph (#vertices) and numbers of edges with positive (#positive edges) and negative (#negative edges) weights are shown.
(DOCX)

**S11 Table. Coevolution of surface sites of COX1 and interactions with other proteins of the respiratory complex IV.** The protein surface sites, sites contacting with other subunits of the complex and noncontact interface sites are identified as described in Methods. The following definitions of three sets of sites participating in interactions with other subunits of the protein complex are considered: (i) sites forming direct contacts (CONT) with other mitochondrially-encoded subunits of COX, (ii) sites forming direct contacts (CONT) with nuclearly-encoded subunits and (iii) the union of sites forming contacts with any other subunit and noncontact interface sites (CONT + ENC_interface). The complementary subsets of sites are noncontact surface sites (NON CONTACT) and surface noncontact noninterface sites (ENC_noninterface). For each set of interacting sites (i-iii) two contingency tables with distributions of observed and expected counts of sites in coevolving groups are shown. The expected counts were obtained by sampling random subgraphs having the same or higher density of edges as were in subgraphs of the contact graph which correspond to the sets (i-iii) of interacting sites. For each group of coevolving sites the Jaccard-index is used as a measure of its overlap with sets of interacting sites (i-iii), for each set two p-values were calculated: the fraction of samples having the same or greater statistics as observed (upper p-value) and the fraction of samples having the same or smaller statistics (lower value). Low values of the upper (lower) p-values correspond to enrichment (avoidance) of sites addressed into each group of coevolving sites among sites interacting with other subunits (i-iii). For each set of interacting sites (i-iii), the $hi^2$ statistic is used as a measure of deviations of observed site counts from expected ones for all groups together, for which a "table p-value" is calculated.
(DOCX)

**S12 Table. Coevolution of surface sites of COX2 and interactions with other proteins of the respiratory complex IV.** See S11 Table for detailed description.
(DOCX)

**S13 Table. Coevolution of surface sites of COX3 and interactions with other proteins of the respiratory complex IV.** See S11 Table for detailed description.
(DOCX)

**S14 Table. Coevolution of surface sites of CYTB and interactions with other proteins of the respiratory complex III.** The protein surface sites, sites contacting other subunits of the complex and noncontact interface sites are identified as described in Methods. The following definitions of two sets of sites participating in interactions with other nuclearly-encoded subunits of the protein complex are considered: (i) sites forming direct contacts (CONT) with other subunits and (ii) the union of sites forming contacts with other subunits and noncontact interface sites (CONT + ENC_interface). The complementary subsets of sites are noncontact surface sites (NON CONTACT) and surface noncontact noninterface sites (ENC_noninterface). For each set of interacting sites (i-ii) two contingency tables with distributions of observed and expected counts of sites in coevolving groups are shown. The expected counts were obtained by sampling random subgraphs having at the same or higher density of edges as were in subgraphs of the contact graph which correspond to the sets (i-ii) of interacting sites. For each group of coevolving sites the Jaccard-index is used as a measure of its overlap with sets of interacting sites (i-ii), for each set two p-values were calculated: the fraction of samples having the same or greater overlap as observed (upper p-value) and the fraction of samples having the same or smaller overlap (lower value). Low values of upper (lower) p-values correspond to enrichment (avoidance) of sites addressed into the corresponding group of coevolving sites among sites interacting with other subunits (i-ii). For each set of interacting sites (i-ii), the hi^2 statistic is used as a measure of deviations of observed site counts from expected ones for all groups together, for which a "table p-value" is calculated.
(DOCX)

**S15 Table. Coevolution of surface sites of ATP6 and interactions with other proteins of the respiratory complex V.** See S14 Table for detailed description.
(DOCX)

**S16 Table. Substitutions rates in groups of coevolving sites have changed during evolution of Metazoa and Fungi.** Each protein has changed substitution rates in groups of coevolving sites several times during evolution, such changes are superimposed on the phylogeny of mitochondrial proteins. For each protein the following statistics are shown: (i) the number of branches on the tree to which episodes of changes of substitution rates have been assigned, for some of these branches the assignment may be ambiguous (see Methods), (ii) a number of these branches for which all five proteins accumulated enough substitutions to test the concordance of changes and (iii) the number of parental branches of (ii) which could be unambiguously used as identifiers of episodes of changes of substitution rates for testing for concordance.
(DOCX)

**S1 Fig. Numbers of predicted concordantly evolved pairs for different nominal p-values in the data, compared to the null distribution.** Black dots indicate data points, boxes with whiskers indicate simulation results. Top and bottom of each box correspond to the 75th and 25th percentile, whiskers correspond to the 95th and 5th percentile.
(TIF)

**S2 Fig. Numbers of predicted discordantly evolved pairs for different nominal p-values in the data, compared to the null distribution.** Black dots indicate data points, boxes with whiskers indicate simulation results. Top and bottom of each box correspond to the 75th and 25th percentile, whiskers correspond to the 95th and 5th percentile.
(TIF)

**S3 Fig. Numbers of predicted concordantly (a, b) and discordantly (c, d) evolved pairs for different nominal p-values in the simulated data, compared to the null distribution.** a,c—episodic positive selection, b,d—hitchhiking. Black dots indicate data points, boxes with whiskers indicate simulation results. Top and bottom of each box correspond to the 75th and 25th percentile, whiskers correspond to the 95th and 5th percentile.
(TIF)

**S4 Fig. Phylogeny for adapting population of suboptimal genotypes, simulated by Santa-Sim.** Since initial genotypes contain many sites with non-optimal amino acids, there are multiple available ways for adaptation, which lead to coexistence of multiple long living clades and causes clonal interference.
(TIF)

**S5 Fig.** Illustrative pairs of sites which evolved under positive (a) or negative (b) epistasis during the forward simulation of evolution. Under positive epistasis, substitutions which were followed (empty green circles) or preceded (green dots) by a substitution at the other site, as well as same-branch substitutions and substitutions which have both leading and trailing counterparts (beige dots) are overrepresented. On the contrary, under negative epistasis, substitutions without immediate leading or trailing counterparts at the other site (black dots) are overrepresented.
(TIF)

**S6 Fig. Pairs of sites of the simulated genome which evolved independently (black dots) or under epistasis (red dots).** The horizontal axis indicates excess (a) or deficit (b) of rapid consecutive substitutions. The vertical axis indicates the excess of clustering (high positive values of clustering z-score) or repulsion (low negative values of clustering z-score) of non-consecutive substitutions for a site pair. Pairs with positive epistatic interactions have elevated values of the association statistics, and pairs with negative interactions, decreased values.
(TIF)

**S7 Fig. Correlation between MAP statistics and association statistics for concordantly evolving site pairs.** Dot color corresponds to the distance between the sites in 3D protein structures: red for contacting sites and black for distant sites.
(TIF)

**S1 Data. The tree, alignments, predicted pairs of epistatically interacting sites and all other data required for analyses mentioned in the Methods section with the detailed description of file contents.**
(ZIP)

**S1 Spreadsheet. Quantification of data shown in S1 Fig.**
(XLSX)

**S2 Spreadsheet. Quantification of data shown in S2 Fig.**
(XLSX)

**S3 Spreadsheet. Quantification of data shown in S3 Fig.**
(XLSX)

**S4 Spreadsheet. Quantification of data shown in S6 Fig.**
(XLSX)

**S5 Spreadsheet. Quantification of data shown in Fig 3.**
(XLSX)

**S6 Spreadsheet. Quantification of data shown in S7 Fig.**
(XLSX)

**S7 Spreadsheet. Quantification of data shown in Fig 6.**
(XLSX)

## Author Contributions

**Conceptualization:** Georgii A. Bazykin.

**Formal analysis:** Alexey D. Neverov, Anfisa V. Popova, Gennady G. Fedonin, Galya V. Klink.

**Investigation:** Alexey D. Neverov, Anfisa V. Popova, Gennady G. Fedonin, Galya V. Klink.

**Methodology:** Alexey D. Neverov, Georgii A. Bazykin.

**Software:** Alexey D. Neverov, Anfisa V. Popova, Gennady G. Fedonin, Evgeny A. Cheremukhin.

**Visualization:** Anfisa V. Popova.

**Writing – original draft:** Alexey D. Neverov, Georgii A. Bazykin.

**Writing – review & editing:** Alexey D. Neverov, Anfisa V. Popova, Georgii A. Bazykin.

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
