## [Decision Letter · Decision Letter 0]

27 Mar 2020

Dear Dr Neverov,

Thank you very much for submitting your Research Article entitled 'Episodic evolution of coadapted sets of amino acid sites in mitochondrial proteins' to PLOS Genetics. Your manuscript was fully evaluated at the editorial level and by independent peer reviewers. The reviewers appreciated the attention to an important problem, but raised some substantial concerns about the current manuscript. Based on the reviews, we will not be able to accept this version of the manuscript, but we would be willing to review again a much-revised version. We cannot, of course, promise publication at that time.

If you decide to revise the manuscript for further consideration at PLOS Genetics, please aim to resubmit within the next 60 days, unless it will take extra time to address the concerns of the reviewers, in which case we would appreciate an expected resubmission date by email to plosgenetics@plos.org.

[LINK]

We are sorry that we cannot be more positive about your manuscript at this stage. Please do not hesitate to contact us if you have any concerns or questions.

Yours sincerely,

Jianzhi Zhang

Associate Editor

PLOS Genetics

Kirsten Bomblies

Section Editor: Evolution

PLOS Genetics

Reviewer's Responses to Questions

**Comments to the Authors:**

Reviewer #1: In this manuscript, the authors modified an algorithm from a previous study, to infer pairs of amino acid sites that experienced excessive or deficient amount of correlated substitutions, along phylogenetic trees for mitochondrial proteins. They further found that concordant sites are spatially close to each other while discordant sites are far away on protein structures. The authors then clustered sites into coevolving groups and indicated that some groups are related to protein surface interaction, and the burst of substitutions for groups happen at some branches more than the others among different OXPHOS proteins.

The pattern and mechanism of site coevolution in protein have been of wide interest. While this study conducted detailed analyses, I recommend the authors discuss more about the comparison between existing studies and their findings, to reflect the novelty of their work. For example, Levin and Mishmar published a study in 2017 on Nature Ecology and Evolution also on OXPHOS proteins and other genes, discovering extensive compensatory substitutions happening on related branches of large phylogeny, affecting amino acid sites within a close physical distance with each other in 3D structure. This should be cited and discussed.

Below are detailed comments.

Line 25. “mitochondrially encoded” should be “mitochondrion-encoded”.

Line 157. The authors claim that “At most of the negatively associated site pairs, the substitutions at the two considered sites tended to occur in lineages that were not only distinct, but also remote from each other on the phylogeny”. Is it supported by statistical test, i.e. such substitutions are more distant on the tree than expected?

Line 159. For both table 1 and table 2, instead of a table showing number of sites at different P-value cutoff, a histogram showing the distribution of P-values of all sites might provide more information on the false discovery. Also, there should be multiple testing correction for FER.

Line 211. Is the “spurious” correlation controlled here referring to the analysis at Line 443? If yes, this might need to be clarified in the main text; if not, what is the criteria for spurious here?

Line 224. In Table 5, the group 4 of atp6 contain only 1 site. Is this a valid coevolving group?

Table 5 has no multiple testing correction for the P-value despite large number of statistical tests. Also, the authors didn’t mention by what test the p-values were derived.

Line 243. For P<0.05 threshold used here, no multiple testing correction is conducted.

Line 256. “APT6” should be “ATP6”

Line 280 “where” should be “were”

Line 435. “in a random dataset”: how is this random dataset constructed?

Line 507. “chi^2” should be “chi square”

In Figure legends, panel index (a, b, c, etc) are inconsistent with those in the Figure (A/B/C…)

Line 851. second “left” might be “right”

Line 861. Figure 4 does not have panel d, e, f.

Line 870. “… function of ATP synthase” should be “… function of ATP6 in ATP synthase”

Reviewer #2: attached

Reviewer #3: This work builds on previous studies from this group (refs 36 and 37) in detecting epistasis based on the timing of mutational events. The authors include this approach to detect positive interactions, and further extend these analyses to detect negative epistatic interactions.

An important feature of this paper's approach is that instead of using time-resolved evolution of influenza sequences, they use 5 OXPHOS pathway proteins. OXPHOS is a classic model where phylogenetic, structural and cellular biology can be used to understand epistasis and co-evolution. Thus another defining feature of this study is that while the earlier influenza model evolved over a microevolutionary timescale of ~ 30 years, OXPHOS evolution occurred over a macroevolutionary timescale.

While I find the new approach and model intriguing, it was difficult for me to follow exactly what analyses the authors performed, in particular how the statistics are calculated, especially this new negative epistatic statistic. Part of this is because the authors simply refer back to their earlier publications for methodological and even broader ideas, expecting the reader to also read those papers in depth.

Thus the impact of this paper depends heavily upon the strengths and frailties of this group's earlier publications, a kind of "literature epistasis" which could be avoided by being more direct and explicit in describing their methods and broader context of the current study.

As a result I unfortunately cannot really comment on the quality of the analysis or conclusions of the study, especially the new model of detecting negative epistasis. I would recommend a heavy rewrite clarifying methodology, including better qualitative descriptions of what was done, quantitative descriptions (equations), more clear null models, and the sharing of code, at least for calculation of their "statistics".

I also think that at the core of this methodology is an assumption that the timing of mutational events reflects epistasis, when it might simply reflect a coevolutionary processes which could include additive or epistatic interactions. Orthogonal analyses, including generative null models which include different scenarios of epistasis, might more convincingly support their conclusions that the observed phenomenon is epistasis per se.

Finally, I think the paper would benefit throughout, in figures and text, from clarification of the major discoveries made using this new approach and model system, including clarification of the "LEGO" analogy, discussion of what new insights negative epistasis provided, and the power of using phylogeny in large protein families which span long evolutionary distances.

Major comments

1) Please clarify methods and share code.

The methods section requires an extensive rewrite to a far greater depth than it is at present.

Two major points:

A) In the text there are two usages of the word "statistic": an "association statistic" and an "epistatic statistic". Neither of these are described quantitatively (or clearly qualitatively, really) anywhere in the text. It is not OK to ask the reader to read reference 36 and guess where its methods apply here.

This problem snowballs later on, for example in interpretation of their "pseudo-correlation" which is the "sum of epistatic statistics for a site" normalized by the max if positive and min if negative (??), which seems unconventional to me given the potential for massive scale variation between sites.

I gave up here because I couldn't really follow the most basic analyses they performed.

B) In the text the authors mention a "null" however I cannot find a description of how this was generated. The earlier work from this group benefitted from an extensive methods section and null distribution produced by a generative model as well as a data permutation approach. Do establish that they are detecting epistasis, and not simply signals of coevolution, I believe that these null models should include specific models of epistasis, and whether their signatures should be detectable in simulated datasets.

I mention other points at the bottom of the text here in the "Methods comments" section.

Finally, it's great the authors included the data, especially the alignments and trees, but in the case of a computational study like this, where much of the data were generated by computer programs, the code used to generate the data is critical to interpret and replicate the work. It is increasingly unacceptable to publish something like this today without sharing the code used to support the conclusions.

2) Validity of the 'inferring epistasis via timing' approach.

I feel the authors should indicate whether they can detect epistasis per se, and include an analysis of whether different types of epistatic interactions can be detected using their approach.

First point - Is it epistasis?

I believe the authors should more carefully consider the assumption that mutations which co-occur (or do not co-occur) in time happen due to epistasis. This is because co-evolution does not require epistasis. Epistasis means that the _effect_ of a mutation depends upon other genetic sequences. So, one could have co-evolving sites which additively sum to make a fit phenotype and thus do not have any dependency in their effects (see for example additive coevolution at linked sites recently from Rockman https://onlinelibrary.wiley.com/doi/full/10.1002/evl3.139).

Thus, simply because two sequences are observed to change together in a coordinated fashion does not mean that one of the mutations' effects depended on or was altered by the other - they could combine in an additive manner. Similarly, just because two deleterious mutations are seen to not combine together does not mean that their combined product is worse than one would expect alone - it's just that two bad mutations are additively bad. Thus it is possible that the authors' "epistasis" scores simply detect coevolution alone.

Unless I am missing an important detail in these analyses, I think this remains an outstanding problem with the authors' approach. If I understand the generative "Simulations without epistasis" null model presented in reference 36, different types of genetic interactions (additive, negative, positive, suppressing interactions) were not explicitly modeled, and therefore we do not know if we can distinguish epistasis from simple co-evolutionary signal that might include additive and epistatic effects.

Second point - "Negative epistasis"

As the authors point out in their final paragraph (line 395) "negative epistasis has, to our knowledge, not been reported from similar data, probably because detection of a deficit of events is harder than detection of an excess"

I cannot comment on the negative epistasis score they present here because its calculation was not described. However, drift and demography seem like the most likely reasons that one would get spurious signals of mutual exclusivity. I hope in the future that the authors address this directly in the text and with figures and code.

In sum, I think the paper, and growing body of work presented by this group in relation to how the timing of mutational events informs about coevolution, would benefit from a deeper discussion of different quantitative genetic and evolutionary scenarios where this method is strong and could fail. Simulation of the evolutionary process under different models of the effects of mutation combinations might be required to address this problem. For these, not only the branching process but also the type of genetic interaction (additive, negative, positive, suppressing interactions) would benefit the analyses.

Minor comments

- How do the authors independently support their ability to detect negative epistatic interactions? For example, in the case of negative epistasis between a and b positions, when deleterious mutation a -> A occurs, the sequence at another position b becomes more essential than it was before because it is less tolerant to other b -> B mutations. If this were the case, should it lead to a signal of purifying selection at b (and linked/coevolving sites?) in 'A' lineages compared to 'a' lineages? Or am I describing the analysis itself here?

- Why leave out leading and trailing sites in the analysis (line 422)? The "directional epistasis" described earlier by this group was impressive because it implied a true epistatic interaction. I don't think the authors have formalized the argument, but a two-locus suppressing relationship, where mutation at locus a -> A is deleterious, but rescued by otherwise neutral b -> B mutation, would lead to ab -> aB -> AB genotype orders but not a -> aB -> AB in phylogenies. (other explanations could be either changing background mutations which alter B's effect, drift, or environmental change).

Apart from drift, for influenza, the suppressing 2-locus landscape or changing environment hypotheses seem most likely, because it seems unlikely that the genetic environment could change so dramatically over periods of < 30 years. However, over hundreds of thousands to millions of years as in the OXPHOS dataset, perhaps other scenarios are more likely?

- For macroevolutionary timescales such as the OXPHOS system, detecting epistasis by "timing" might be prone to problematic inferences due to a process similar to incomplete lineage sorting, whereby rapidly evolving (highly diverse) ancestral sequences might lead to sequences in divergent branches which appeared to independently evolve when in fact these sequences were present at low frequencies in ancestral populations prior to divergence. Some of the authors have previously worked with a similar issue in studying homoplasy. Was this a likely problem in the current study, and did the authors address it directly?

- The earlier study benefitted from having time-resolved measurements of genetic changes in influenza. With these data they could detect "directionally epistatic" interactions in historical influenza sequence datasets, suggesting epistatic interactions might be important over the timescales of decades (see below). With the OXPHOS system, we are considering hundreds of thousands of years of divergence. So we sometimes might not be as confident about the timing of different ancestral branch points. Does this affect the analyses in the present study?

Further, as the authors pointed out here and in reference 36, by having large genetic distances, mutual information or DCA approaches work well for detecting co-evolving sites because mutation diversity is less affected by phylogeny. How did phylogeny help compared to these approaches? I don't really see how the approach presented in this study has led to different conclusions compared to other approaches.

- The authors refer to LEGO blocks in the author summary. I love LEGOs, especially when used as an analogy outside the tired context of the "modularity" of cellular signaling and metabolic networks, and so obviously this analogy piqued my interest. Unfortunately, I felt this interesting idea needed to be elaborated on more clearly in this explanatory paragraph as well as in the text.

- Tables. I have never seen tables like this. Most tables appear to be count data across parameter scans, with columns corresponding to arbitrary cutoffs in pre-FDR p-values. Next to each count statistic is an extra "FDR" column, which adds to confusion and makes the tables even more difficult to follow. The counts within a column can differ by orders of magnitude and it isn't clear if this is because there are simply more site pairs for the given protein or whether it is biologically meaningful. For these tables, I encourage the authors to consider whether they are really benefitting their claims, and if so, then to consider showing counts as proportions of total pairs and plotting these values as a function of FDR. The pre-multiple testing p-value is not informative.

- Figures 1 and 2. It seems these are meant to illustrate how the algorithm works. Not much useful data can be gleaned from the figures, which include thousands of leaves, and comparison between right and left-handed mirror trees is difficult to follow at first. It was much clearer how the algorithm worked in the simpler graphs from earlier publications from these authors (Plos Gen refs 36 and 37).

- Figure 3. Big issue: what is that color statistic? It's a proportion - why not put it in log scale so that it is proportionally the same for values < 1 as for values > 1? Why change the range of values for each panel? Seems there should be a standard scale at least down columns, where the same statistic has been calculated across proteins. But then, indeed, the values vary wildly column-wise and row-wise between panels. Why is this? More explicit discussion of why these values vary so dramatically would help.

- Figure 3. Why does cluster 6 in panel (C) or cluster 4 in (J) ... etc ... not connect to itself?

- Consider annotating your figures with a bit more text. For example, the panels of Figure 3 are so interrelated by rows and columns such that it is really one figure with different facets. Labelling the rows and columns would help the eye, as well as help understand what statistic is that is being mapped on the figure.

- Similarly, for Figure 4, label the panels with the name of the protein in the structure, and put an axis with a "180º" arrow around it to indicate that rotated the structure is rotated.

- Each of the proteins studied in the work has a crystal structure and the authors report contact graphs for all of them in figure 3. Why not show all the structures as in figure 4, at least in the supplement?

- For the data please use a more standard compression algorithm than '7z'. Your readers shouldn't have to search the internet to download special software to open arcane filetypes to simply see the data.

Methods comments

Line 30: "Evolution in most protein sites is constrained by alleles in other sites, this phenomena is called epistasis."

A set of additive interactions between coevolving sites could give rise to such apparent constraints.

Line 413: 3 structures are mentioned when structural data for 5 proteins are described in the text. Why only 3 and not 5? The information about the B. taurus alignment is confusing as well. Please rephrase to more explicitly describe what you did there.

Line 421: "Mutations that followed one another had higher weight" - what was the equation for that? What was the exact exponential penalty? What were the time units for this statistic?

Line 422:

"Unlike [36], we did not distinguish between “leading” and “trailing” sites; instead, the epistatic statistic was defined for an unordered pair of sites as the sum of the statistics for the two corresponding ordered pairs."

Why this change? Did the authors find weak evidence for this directional epistasis?

**Have all data underlying the figures and results presented in the manuscript been provided?**

Reviewer #1: Yes

Reviewer #2: Yes

Reviewer #3: None

PLOS authors have the option to publish the peer review history of their article (what does this mean?). If published, this will include your full peer review and any attached files.

Reviewer #1: No

Reviewer #2: Yes: Daniel Lyons

Reviewer #3: No

---

## [Decision Letter · Decision Letter 1]

2 Sep 2020

Dear Dr Neverov,

Thank you very much for submitting your Research Article entitled 'Episodic evolution of coadapted sets of amino acid sites in mitochondrial proteins' to PLOS Genetics. Your manuscript was fully evaluated at the editorial level and by the three original peer reviewers. The reviewers appreciated the attention to an important topic but identified some aspects of the manuscript that should be improved.

We therefore ask you to modify the manuscript according to the review recommendations before we can consider your manuscript for acceptance. Your revisions should address the specific points made by each reviewer.

[LINK]

Yours sincerely,

Jianzhi Zhang

Associate Editor

PLOS Genetics

Kirsten Bomblies

Section Editor: Evolution

PLOS Genetics

Reviewer's Responses to Questions

**Comments to the Authors:**

Reviewer #1: The authors have properly addressed the reviewers’ comments and substantially improved the manuscript from the last version.

The panel names for Figure 5 should be a and b rather than cd, or Fig 4 and Fig 5 should be merged together.

Reviewer #2: Attached

Reviewer #3: I like the new version of the manuscript more and the authors have addressed most of my concerns. I appreciate the effort they put into their response to my comments and find the analysis of comparison between DCA and their method very interesting.

However, I still have concerns about the authors' (quantitative) null prediction for evolution under no epistasis. I do not understand why the selection process of epistasis cannot be modeled to confirm their reasoning about what processes should give rise to observed "repelling" or co-occurrence patterns. Correspondingly, sections (table 2, e.g.) where these hypotheses are discussed do not seem to have much theoretical (analytic or numerical simulation) support; no quantitative, concrete hypothesis of the null distribution of substitutions under epistatic vs non-epistatic selection seems to have been tested. For example, if section beginning line 164 was such a test, why were no epistatic parameters added to illustrate that these were necessary and sufficient to cause repelling/co-occuring patterns of evolution? The lack of these models makes a lot of their reasoning around Line 207 fuzzy to me.

Line 56-57: "The main factor affecting the fixation probability is selection favoring some variants over others." Please clarify - the strength of selective pressure relative to mutation rate, demography, and drift are what affect fixation probability

Section beginning line 84. Please define epistasis (it was done briefly in the "author summary" section but must be elaborated in the main text).

Line 94. Evidence was presented for varying selective pressures in protein evolution. No evidence of epistasis was presented in the paragraph above and so you can't say "These data suggest that the effect of epistasis on the rate of evolution strongly depends on the identity of the interacting partners"

Line 97. This is good evidence of epistasis in these proteins. Is there really nothing else?

Line 99. First mention of OXPHOS and you have never formally present which proteins you are studying in the text.

Line 107. Coevolution can occur in the absence of epistasis and therefore is not evidence of epistasis.

Table 1. # pairs is shown. How many pairs were tested?

Line 141: 'Statistic' - please explain what this statistic is, and whether and how this is different from the "association statistic"

Line 148: "the test has different power to detect positive and negative associations." Providing an intuition of why this is the case will likely help the reader to understand the tests that have been performed.

Line 153: FER vs FDR. Please provide a discussion somewhere in the text on how to interpret the FER as you provided to one of the reviewers.

Section line 164: Interesting that no simulation under different demographic scenarios could produce the patterns of substitution you observe in the OXPHOS proteins. Are these methods sufficient to produce any such a signal at all? I.e., what parameters must you change or introduce to reproduce the patterns of substitution you observe in OXPHOS?

This section would benefit from more explicit statements of hypotheses being tested and ruled out.

Line 207: Clear hypotheses are presented for epistasis of positive or negative nature. However, without clear analytical or numerical demonstration of how these signals would emerge as part of the evolutionary process, it is not clear whether there is a quantitative prediction of the null hypothesis.

Line 258: "To explain it, one needs to invoke an episode of selection favoring substitutions at both sites individually, but disfavouring their combination – i.e., an episode of negatively epistatic selection." This seems to be a quantitative prediction which must be defended with analytically or numerically.

Table 2. Categorization of different types of 'dis/concordant' evolution across episodes of selection are presented. Are the X's and Check Marks in the boxes made by eyeing the data and deciding what fell where? How are there no error or counts here?

line 314. Table S5. Correlation coefficients are significant but small (avg ~0.15). Please state the range of the statistics here and wherever there are mentions of 'significance' pointing to the supplement in the main text so the reader has a sense for the scale of these tests.

**Have all data underlying the figures and results presented in the manuscript been provided?**

Reviewer #1: Yes

Reviewer #2: Yes

Reviewer #3: Yes

PLOS authors have the option to publish the peer review history of their article (what does this mean?). If published, this will include your full peer review and any attached files.

Reviewer #1: No

Reviewer #2: **Yes: **Daniel Lyons

Reviewer #3: No

---

## [Editor Report · Decision Letter 2]

7 Dec 2020

Dear Dr Neverov,

We are pleased to inform you that your manuscript entitled "Episodic evolution of coadapted sets of amino acid sites in mitochondrial proteins" has been editorially accepted for publication in PLOS Genetics. Congratulations!

Yours sincerely,

Jianzhi Zhang

Associate Editor

PLOS Genetics

Kirsten Bomblies

Section Editor: Evolution

PLOS Genetics

Comments from the reviewers (if applicable):

**Data Deposition**

http://datadryad.org/submit?journalID=pgenetics&manu=PGENETICS-D-20-00324R2

**Press Queries**

---

## [Editor Report · Acceptance letter]

18 Jan 2021

PGENETICS-D-20-00324R2 

Episodic evolution of coadapted sets of amino acid sites in mitochondrial proteins 

Dear Dr Neverov, 

We are pleased to inform you that your manuscript entitled "Episodic evolution of coadapted sets of amino acid sites in mitochondrial proteins" has been formally accepted for publication in PLOS Genetics! Your manuscript is now with our production department and you will be notified of the publication date in due course.

With kind regards,

Melanie Wincott

PLOS Genetics

On behalf of:
